# Meteorological, snow and soil data (2013-2019) from a herb tundra permafrost site at Bylot Island, Canadian high Arctic, for driving and testing snow and land surface models

Florent Domine[1,2,3], Georg Lackner[1,2,4], Denis Sarrazin[2], Mathilde Poirier[2,5], Maria Belke-Brea[1,2,6]

[1] Takuvik Joint International Laboratory, Université Laval (Canada) and CNRS-INSU (France), Québec, Canada
[2] Centre d'Études Nordiques, Université Laval, Québec, Canada
[3] Département de chimie, Université Laval, Québec, Canada
[4] Département de génie civil et de génie des eaux, Université Laval, Québec, Canada
[5] Département de biologie, Université Laval, Québec, Canada
[6] Département de géographie, Université Laval, Québec, Canada

*Correspondence to*: Florent Domine (florent.domine@gmail.com)

**Abstract.** Seasonal snow covers Arctic lands 6 to 10 months of the year and is therefore an essential element of the Arctic geosphere and biosphere. Yet, even the most sophisticated snow physics models are not able to simulate fundamental physical properties of Arctic snowpacks such as density, thermal conductivity and specific surface area. The development of improved snow models is in progress but testing requires detailed driving and validation data for high Arctic herb tundra sites, which are presently not available. We present 6 years of such data for an ice-wedge polygonal site in the Canadian high Arctic, in Qarlikturvik valley on Bylot Island at 73.15°N. The site is on herb tundra with no erect vegetation and thick permafrost. Detailed soil properties are provided. Driving data are comprised of air temperature, air relative and specific humidity, wind speed, short wave and long wave downwelling radiation, atmospheric pressure and precipitation. Validation data include time series of snow depth, shortwave and longwave upwelling radiation, surface temperature, snow temperature profiles, soil temperature and water content profiles at five depths, snow thermal conductivity at three heights and soil thermal conductivity at 10 cm depth. Field campaigns in mid-May for 5 of the 6 years of interest provided spatially-averaged snow depths and vertical profiles of snow density and specific surface area in the polygon of interest and at other spots in the valley. Data are available at https://doi.org/10.5885/45693CE-02685A5200DD4C38 (Domine et al., 2021). Data files will be updated as more years of data become available.

## 1 Introduction

The seasonal snowpack covers high latitude regions at low elevation six to ten months of the year (Connolly et al., 2019). Snow physical properties such as specific surface area (SSA) and density determine albedo (Carmagnola et al., 2013), an essential component of the surface energy budget. Snow thermal conductivity determines heat exchanges between the atmosphere and the ground and therefore impacts the permafrost thermal regime (Zhang, 2005;Barrere et al., 2017). Where

permafrost is absent, snow thermal conductivity determines whether and when the ground freezes, with very strong impact on nutrient recycling and the accumulation of organic compounds in soils (Saccone et al., 2013;Myers-Smith and Hik, 2013;Buckeridge and Grogan, 2008). Snow thermal conductivity also strongly influences surface air temperature (Domine et al., 2019) and inadequate simulations of this variable can modify simulated air temperature by up to 4°C, potentially affecting

meteorological models and high latitude weather forecasts. Lastly, snow physical properties affect wildlife. For example, lemming, small rodents of the high Arctic, live, move, feed and reproduce under the snow nine months of the year in the high Arctic (Poirier et al., 2019;Bilodeau et al., 2013). Their population cycles have intrigued scientists for decades (Fauteux et al., 2015) and recent studies have indicated that snow physical properties, and in particular the hardness of the snow basal layer, may strongly impact lemming reproductive success in winter and their summer population dynamics (Domine et al., 2018b).

Likewise, larger arctic herbivores such as caribou are strongly affected by snow physical properties, which determines their access to food under the snow (Langlois et al., 2017).

Adequately simulating snow physical properties is therefore essential for understanding and/or projecting climate, meteorology, the state of permafrost, nutrient cycling and carbon storage in soils and therefore vegetation growth and the carbon budget of Arctic soils and permafrost, and wildlife ecology and population dynamics. Despite these critically high

stakes, there is today no detailed snow physics model capable of simulating Arctic snowpack physical properties adequately. (Domine et al., 2016b) have shown that both detailed snow models Crocus (Vionnet et al., 2012) and SNOWPACK (Bartelt and Lehning, 2002) failed to simulate essential characteristics of the snowpack at the high Arctic site of Bylot Island (73°N). In particular, simulated snow density profiles were inverted relative to observations. Both models predicted dense basal layers and light top layers while most snow observations in the high Arctic have reported low-density basal layers made of depth

hoar and high-density upper wind slabs (Domine et al., 2016b;Derksen et al., 2009;Domine et al., 2002;Domine et al., 2012;Gouttevin et al., 2018). The explanation proposed (Domine et al., 2019;Domine et al., 2016b) is that Crocus and SNOWPACK were calibrated using data from sites in the Alps, i.e. for mid latitude warm thick snowpacks while the Arctic features cold thin snowpacks. In the Alps, an essential driving variable in snowpack vertical profiles of physical properties is compaction by the snow overburden. In the thin Arctic snowpack, this process is negligible and the main process involved in

determining the evolution of the density profile after precipitation and wind compaction is the upward flux of water vapor, caused mostly by convection within the snowpack (Trabant and Benson, 1972;Sturm and Benson, 1997;Sturm and Johnson, 1991;Domine et al., 2016b). This flux is driven by the large vertical temperature gradient which redistributes mass from lower to upper layers. This process is so intense and the associated mass loss so large that it sometimes leads to the collapse of the basal depth hoar layer (Domine et al., 2016b), even in the low Arctic (Domine et al., 2015). This process is not simulated by

Crocus or SNOWPACK, leading to erroneous outputs.

Upward water vapor fluxes are also the main determinant of snowpack vertical profiles of physical properties in many areas of the boreal forest (Sturm and Benson, 1997). Since Arctic and boreal forest snowpacks together represent by far the most important seasonal snowpacks on Earth on an areal basis (Sturm et al., 1995), it is essential that data sets be available, which allow the testing of snow models and their application to high latitudes. At present, there is not to our knowledge any multi-

year high Arctic data set complete enough to drive and validate in detail snow physics model. The northernmost site used in the latest snow model intercomparison project (SNOWMIP, (Krinner et al., 2018)) is Sodankylä, Finland, 67°N. Although it is classified as "Arctic" in (Krinner et al., 2018), Sodankylä is in the boreal forest, while Arctic usually refers to regions above tree-line. In the boreal forest, the dense wind slabs observed in the Arctic do not form and snow properties are markedly different from those on Arctic tundra (Sturm et al., 1995). (Boike et al., 2018) have provided a 20-year data set of permafrost,

active layer and meteorological data for a site near Ny-Ålesund, Svalbard (N:78.5, E:11.6) suitable for driving land surface and snow models. However, while this data set can be used for numerous valuable applications, the snow validating data are limited to snow depth and to snow pit observations in late April or early May. The snow physical data is comprised of density at several heights and of the vertical temperature profile when the pit was dug. These data are useful but are probably not sufficient for thoroughly testing snow physics model performance under Arctic conditions. (Boike et al., 2019) also presented

a 16-year data set of permafrost, active-layer, and meteorological conditions for Samoylov Island in Siberia (N:72.3, E:126.5). The site is in the Lena river delta and features ice-wedge polygons, with a very high fraction of ground ice. The permafrost data, together with a previous paper (Boike et al., 2013) are extremely detailed, so that this data set is certainly particularly useful for permafrost simulations. Regarding snow however, data are more limited and comprised of snow depth, time-lapse photographs, and some spring measurements of snow properties such as density and thermal conductivity. Snow precipitation

has not been measured there.

    The Samyolov site has been used to test the SNOWPACK model. (Gouttevin et al., 2018) used a one-year driving data set to simulate snow and used snow pit data from a field campaign in April, as well as ground temperature monitoring at 5 cm depth as validating data. They modified the SNOWPACK model to adapt it to Arctic conditions and in particular modified grain-growth rate laws. They however did not treat upward water vapor fluxes explicitly. That study constitutes valuable progress

towards the elaboration of an Arctic snow model, but a one-year test is not sufficient to oversee the variety of high Arctic conditions. For example, in their study, they encountered high density "indurated" depth hoar which is frequent but far from ubiquitous in the high Arctic. The motivation of the present work is therefore to provide over an extended period driving and testing data for snow physics at a high Arctic site, so that the ability of snow physics and land surface schemes can be tested in a variety of meteorological situations in these high Arctic conditions.

We provide standard meteorological data for driving models: air temperature and relative humidity, atmospheric pressure, wind speed, short wave (SW↓) and longwave (LW↓) incoming radiation, and precipitation. Detailed soil properties such as density, granulometry, organic carbon content and thermal conductivity at several depths are also provided. For validation, we provide continuous monitoring of snow depth, upwelling shortwave radiation (SW↑), surface albedo, upwelling longwave radiation (LW↑), surface temperature, snow temperature at five heights, snow thermal conductivity at three heights, soil

temperature and volume liquid water content at five depths and soil thermal conductivity at 10 cm depth. The data cover a 6-year span from 11 July 2013 to 25 June 2019. Data from 2019-2020 could not be retrieved due to the COVID-19 pandemic, which prevented access to our site. However, data from future years will be added to the set as they become available. Furthermore, field campaigns at this site were possible in May 2014, 2015, 2017, 2018 and 2019 and we also present snowpit

observations of snow stratigraphy and measurements of vertical profiles of snow density and SSA for those years. In May 2016, logistical difficulties prevented access to the site.

## 2 Study site and instruments

Our study site is in Qarlikturvik valley on Bylot Island, North of Baffin Island in the Canadian Archipelago (Figure 1). The nearest community is Pond Inlet on Baffin Island, 85 km to the south-east, from which our permanent camp can be accessed by helicopter in summer or snowmobile in spring. Aircraft landing on skis is occasionally possible in spring or on a nearby beach in summer with tundra tyres. Atmospheric monitoring was initiated in August 1993 at the Camp Lake site (Hereafter CAMP, N:73.1567°; W:79.9571°) and air temperature and humidity as well as wind speed (3 m) data are available since that date (CEN, 2020). In July 2004 a 10 m tower (called SILA) was built at (N:73.1522°; W:79.9886°), 1150 m W-SW of CAMP and equipped to measure wind speed and direction (10 m) and air temperature. Data are also available (CEN, 2020) at the same repository and DOI as the CAMP data. The data discussed here are from a comprehensive monitoring station established on 7 July 2013 at the TUNDRA site (N:73.1504°; W:80.0046°), 1700 m to the W-SW of CAMP. The GPS elevation of the site is 20 to 25 m but according to the Canada Atlas maps (atlas.gc.ca), the site is just below the 20 m contour line. Google-Earth indicates an elevation of 25 m. The site has been presented by (Domine et al., 2016b). Briefly, the instruments are within a rather well-drained low-center ice-wedge polygon typical of permafrost landscape. The polygon is about 11 m in its largest dimension and all instruments are within 3 m of its center. Equipment is detailed in Table 1 and includes a tripod supporting meteorological instruments at 2.3 m height, a vertical polyethylene post supporting three TP08 heated needle probes for snow thermal conductivity measurements and another post supporting five thermistors for snow temperature measurements. Below the surface, 5TM sensors from Decagon (now Meter) measure soil temperature and volumetric liquid water content at five depths and a TP08 probe monitors soil thermal conductivity. The instruments are accessed and maintained once a year in summer. Instrument failure thus cannot always be fixed immediately. The CNR4 radiometer was brought back south in 2019 for recalibration by Kipp & Zonen and the drift in sensitivities was considered in data analysis. Some data from 2013-2015 have been reported by (Domine et al., 2016b) and have been used by (Barrere et al., 2017) to simulate snow and ground properties, with driving data presented by (Barrere and Domine, 2017). Most data were recorded by a Campbell CR1000-XT data logger, except soil temperature and volume water content that were recorded by an Em50 logger from Decagon (now Meter). A Reconyx time-lapse camera taking several pictures a day was installed in summer 2016. It was replaced and reoriented in July 2018. Similar cameras at other sites within 3 km were present starting in 2015. In summer 2016, the SALIX meteorological station, fairly similar to the TUNDRA station described here (except there was no CNR4) was deployed 9 km up-valley from TUNDRA, at (N:73.1816°; W:79.7454°). Data from that station were occasionally used for filling data gaps. There is no small-scale topography within the polygon (Figure 1) and in particular no hummock or tussock. The permafrost there has been described by (Fortier and Allard, 2005). It is several hundred meters thick with an active layer 20 to 35 cm deep. The visibly (dark) organic-rich surface soil layer is 2 to 10 cm thick. Soil samples from a vertical profile taken with a vertical

resolution of 5 cm on 3 July 2017 were analyzed for organic carbon content, using the procedure detailed in (Gagnon et al., 2019). We chose a spot where the thawed layer was deepest, 30 cm, within an observed range of 15-30 cm. There was less moss at this spot than in most other places on the polygon. The carbon content decreased from 8.3 to 0.3 kg C per kg of dry soil between 0-5 cm and 20-25 cm depths. A graph is shown in Figure S1, along with soil density and water weight fraction.

Soil granulometry was analyzed as mentioned in (Domine et al., 2016b) for three depth intervals: 0-5 cm, 10-15 cm and 20-25 cm (Figure S2). The 0-5 cm sample was silt loam with a unimodal asymmetric distribution of grain size centered at 51 μm, the 10-15 cm sample was silt loam with a bimodal distribution at 17.4 and 59.0 μm and the 20-25 cm sample was sandy loam with a bimodal distribution at 11.6 and 152.5 μm.

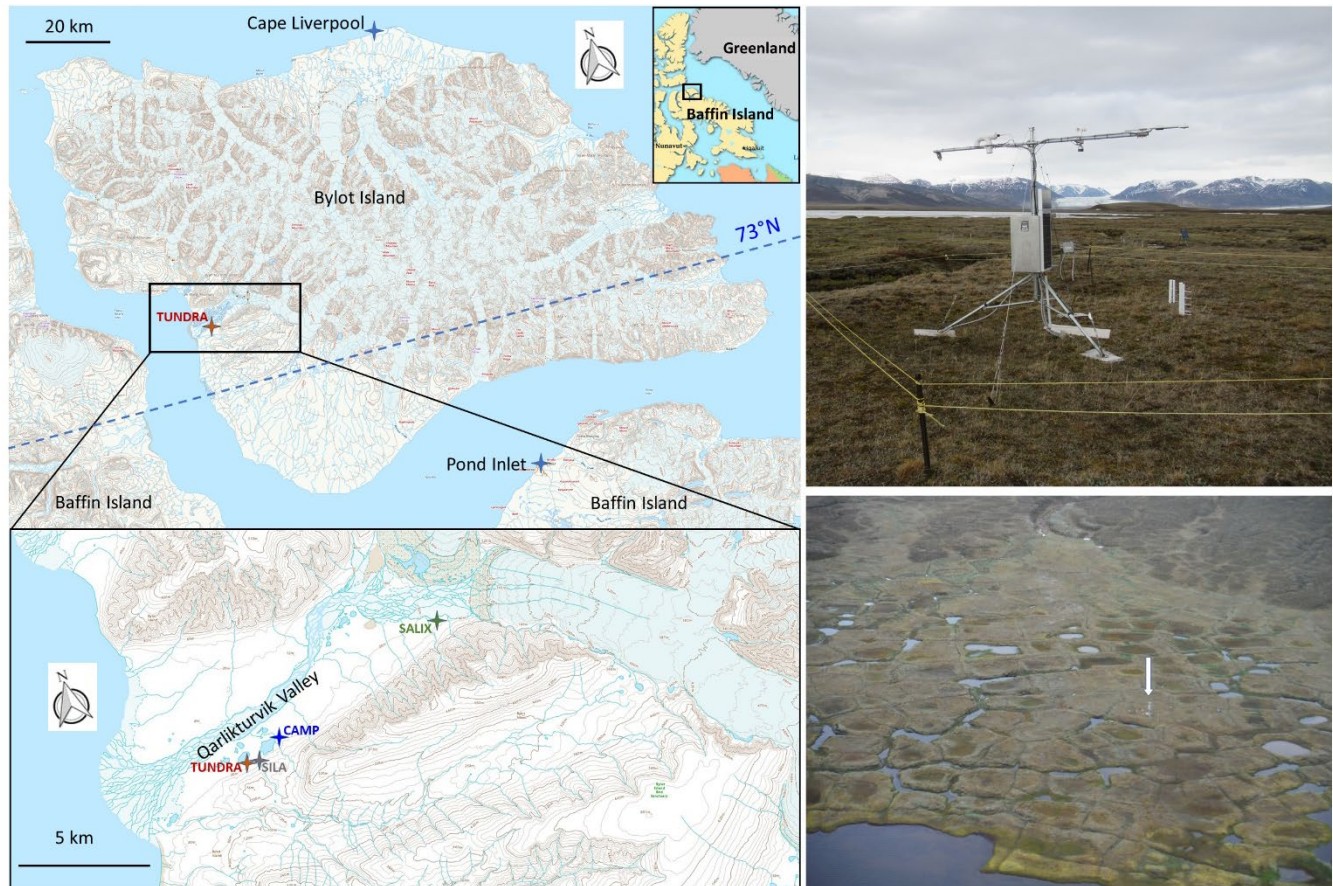

**Figure 1. (Left) map of Qarlikturlik valley on Bylot Island in the Canadian Arctic Archipelago, showing the TUNDRA study site. The ECCC meteorological stations at Cape Liverpool and Pond Inlet, as well as the SILA, CAMP and SALIX stations, all of which were used for data gap filling, are also shown. (Right) view of the polygon where instruments were installed; general aerial view of the polygons area, with arrow pointed at TUNDRA station. Maps from atlas.gc.ca.**

Triplicate measurements of the ground thermal conductivity were performed on 9 July 2013 at 5 cm depth, within 2 m of the

TP08 post, with a TP02 heated needle probe from Hukseflux, yielding values of 0.159 ± 0.004 W m$^{-1}$ K$^{-1}$. The ground temperature was 6.3°C and the fractional water volume content measured with an EC5 probe from Decagon was 0.151.  A

vertical profile of the ground thermal conductivity was measured on 29 June 2014, about 5 m away from the TP08 post, showing values increasing from 0.235 W m$^{-1}$ K$^{-1}$ at 3 cm depth to 1.481 W m$^{-1}$ K$^{-1}$ at 15 cm depth. Another vertical profile of thermal conductivity was measured on 3 July 2017, within 2 m of the 29 June 2014 pit, when soil samples for carbon analysis

were taken. Plots of the vertical profiles, along with the associated profiles of fractional water volume content (obtained with a Decagon EC5 sensor) and temperature, are shown in Figure S3. Significant spatial variability of thermal conductivity is observed, since values near the surface vary by a factor of three within a few meters. Over the years, about 8 soil pits were dug in the TUNDRA polygon for physical measurements, sampling, and instrument installation and maintenance. Variations in soil color and texture were visible to the eye. In particular the thickness of the darker surface soil layer, presumably organic-

rich, varied between 2 and 10 cm. All soil physical variables mentioned here therefore varied within the polygon. More extensive pit digging for extra measurements risked modifying the soil properties in the polygon.

Vegetation in these polygons has been detailed in (Gauthier et al., 1995). It consists mostly of graminoids, sedges and mosses with some prostrate ligneous species: *Salix arctica* and *S. herbacea*. Vegetation height does not exceed 5 to 10 cm, as there is no erect vegetation (Figure 1). The spectral albedo of the site was measured on 11 July 2015 around 17:25 UTC. The sky was

clear but the atmosphere was slightly foggy. The instrument was an SVC spectrometer equipped with an integrating sphere, one Si photodiode detector for the visible and near infrared range and two InGaAs photodiode detectors for the shortwave infrared range. Two spectra were recorded over the 346-2513 nm wavelength range. They are shown in Figure S4 and are essentially identical. The broadband albedo derived from the average of these spectra is 0.18.

In May 2014, 2015, 2017, 2018 and 2019, field measurements were performed in several spots in Qarlikturvik valley. Around

the TUNDRA sites, over 100 measurements of snow depth were performed to obtain a more spatially representative value of snow depth than the one spot measured automatically. Snow pits were dug to observe the stratigraphy and measure vertical profiles of density and SSA. Density was measured by weighing snow sampled with a 100 cm$^3$ box-cutter. SSA was measured by infrared reflectance at 1310 nm using the DUFISSS instrument (Gallet et al., 2009). During each campaign, a pit was dug within 3 m of the thermal conductivity post. Two to seven pits were dug elsewhere in the valley to assess spatial variability.

Logistical difficulties did not allow a field campaign in May 2016.

**Table 1. Instruments used to obtain meteorological, snow and soil data at Bylot Island.**

| Variable | Instrument | Manufacturer | Instrument height/depth | Comment |
|---|---|---|---|---|
| Short-wave radiation down- and up-welling | CNR4 pyranometers, 300-3000 nm with CNF4 ventilator/heater | Kipp & Zonen | 2.3 m | CNF4 on for 5 minutes before hourly measurement |
| Long-wave radiation down- and up-welling | CNR4 pyrgeometers, 4.5 to 42 μm with CNF4 ventilator/heater | Kipp & Zonen | 2.3 m | CNF4 on for 5 minutes before hourly measurement |

| | | | | |
|---|---|---|---|---|
| Snow (winter) or soil (summer) surface temperature | IR 120 infrared sensor, 8 to 14 μm | Campbell Scientific | 1.5 m | Measurement every minute, hourly average recorded |
| Air temperature and relative humidity (relative to liquid or supercooled water) | HC2-S3-XT sensor, inside white PVC tubing, ventilated | Rotronic | 2.3 m | Ventilator on for 5 minutes before hourly measurement |
| Wind speed | Cup anemometer | Vector instruments | 2.3 m | Measurement every minute, hourly average recorded |
| Precipitation | Geonor 200. Complemented with data from Geonor gauges at Pond Inlet and Cape Liverpool. | | 1.5 m | |
| Atmospheric pressure | Not measured, data from Pond Inlet and Cape Liverpool were used. | | | Average value of data from station at Pond Inlet and cape Liverpool |
| Snow depth | SR50A acoustic gauge | Campbell scientific | 2.2 m | Measurement every minute, hourly average recorded |
| Snow thermal conductivity | TP08 heated needle probes | Hukseflux | 7, 17, 27 cm in 2013, changed to 2, 12, 22 cm in July 2014 | Measurement every other day at 5:00 AM local summer time |
| Snow temperature | Pt 100 thermistors | Home-assembled sensors | 2, 7, 17, 27, 37 cm, changed to 0, 5, 15, 25, 35 in July 2018 | Measurement every minute, hourly average recorded |
| Soil thermal conductivity | TP08 heated needle probes | Hukseflux | 10 cm | Measurement every other day at 5:00 AM |
| Soil temperature and volume water content | 5TM sensors | Decagon (now Meter) | 2, 5, 10, 15, 21 cm | Hourly measurement |
| Scenery | Time-lapse camera | Reconyx | 1.5 m | 4 pictures a day |

## 3 Driving data quality check and correction

Several environmental factors and problems with instruments can affect data quality. For example, since the tripod is on permafrost, ground thawing and freezing may modify its leveling which was adjusted in early July every year. However, further shifting can take place later in the summer. Some years, this produced a slight offset in snow depth and in the CNR4 leveling. In winter, frost can build up on the anemometer and block it. Frost can obscure the CNR4 top windows, producing erroneous measurements. All these difficulties were thoroughly investigated and corrected for. In 2016, large surface plates

were placed under each tripod leg, and this significantly reduces tripod movement and tilting. The treatments done to each driving and validating variable are detailed below.

## 3.1 Air temperature

Air temperature was measured with a ventilated HC2-S3-XT sensor at 2.3 meter height. Data from 2013-2014 were missing because of sensor failure but we used CR1000 data logger temperature instead. That sensor was slightly sensitive to radiation. Based on several years of simultaneous temperature measurements of the HCS2-S3-XT and CR1000 sensors, we corrected the CR1000 sensor values by adding a linear function of downwelling SW radiation whose coefficients were optimized to obtain a zero bias. The RMSD was 0.784°C. The sensor was replaced in July 2014 and there was no other data gap. TUNDRA air temperature data were compared to those from the SILA, CAMP and SALIX stations. All differences could be readily explained by topography and basic meteorological concepts, such as katabatic flow at the bottom of the valley which led to colder air at TUNDRA in winter. We are thus confident in the reliability of the air temperature data. The temperature time series is shown in Figure 2.

## 3.2 Relative Humidity

Relative humidity (RH) was also provided by the HC2-S3-XT sensor. This is a Humicap thin film capacitive sensor which provide RH relative to liquid or supercooled water, not ice. It needs to be calibrated and the calibration provided by the manufacturer was checked. We found that for the second sensor, installed in 2014, RH never reached 100% in summer. We therefore multiplied the value obtained by 1.045 so that the 100% value was reached about as frequently as the first year. Regarding winter data, we observed that by plotting RH vs. temperature, maximum values deviated from the ice saturation line (Figure 3). The first sensor gave lower values for the 2013-2014 period while the second sensor gave higher values. We corrected the data so that maximum values for temperature <0°C coincided with the ice saturation values. For the ice and supercooled water saturation vapor pressures, we used the equation of (Huang, 2018). Huang does not mention an accuracy for the supercooled water pressure values, but comparison with measured values available at https://www.engineeringtoolbox.com/water-supercooled-vapor-pressure-d_1910.html (accessed on 5 February 2021) revealed an excellent agreement. At -40°C, the value of Huang was 1.3% higher than the measured value (19.16 vs. 18.91 Pa). At -10°C, the difference is just 0.024% (286.57 vs. 286.50 Pa). In Figure 3, we plotted the ice saturating RH derived from Huang's equations. To make our data fit the ice values, we used equations (1) and (2) for the 2014-2019 and 2013-2014 data, respectively, where T is temperature in Celsius and RH in %.

$$RH_{corrected} = (RH_{measured} \times 1.045)(0.0031 \times T + 0.77) + 21 \qquad (1)$$

$$RH_{corrected} = RH_{measured}(0.00065 \times T + 0.75) + 24 \qquad (2).$$

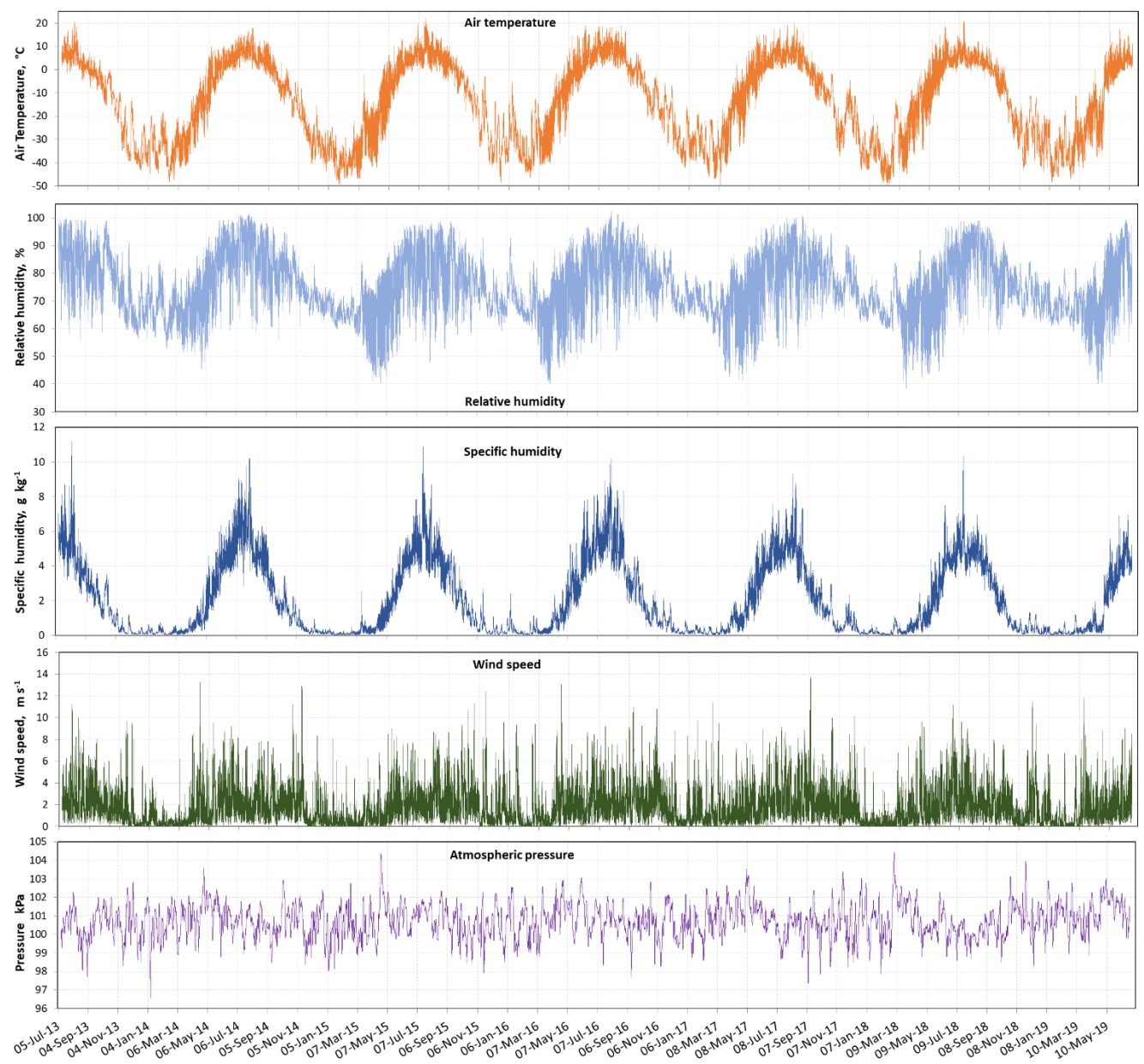

210

**Figure 2. Time series of air temperature, relative and specific humidity, wind speed and atmospheric pressure at the TUNDRA site on Bylot Island.**

### 3.3 Specific humidity

Many models use specific humidity rather than RH relative to water as input variable and we therefore also provide that variable in g of water per kg of moist air. To calculate the partial pressure of water vapor, we used equation (17) of (Huang, 2018). We used PV=nRT for the gas equation of state, where P is pressure, V the volume considered, n the number of moles

in V, R the gas constant and T temperature. Values used in the calculations are 18.01528 g for the molar mass of water, 8.3145 J K$^{-1}$ mol$^{-1}$ R and 28.9647 g for the molar mass of dry air. The humidity time series are shown in Figure 2.

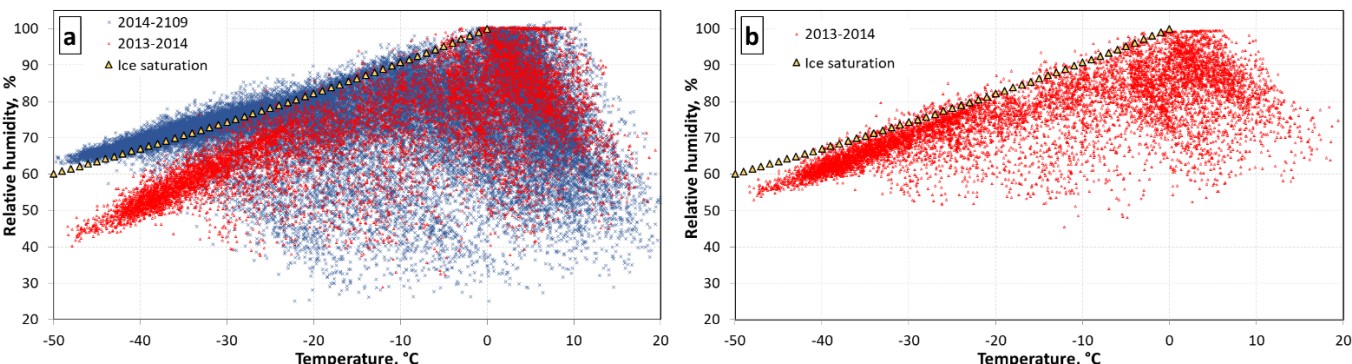

Figure 3. Air humidity relative to liquid water. (a) Uncorrected data from the first RH sensor (red points), used in 2013-2014, and data from the second RH sensor, used in 2014-2019 (blue points), after multiplication by 1.045 to ensure maximum RH reached 100% in summer. The ice saturating RH values relative to supercooled water, based on the equations of (Huang, 2018) are also shown. (b) Corrected data for 2013-2014. Both data sets were corrected with equations (1) or (2) to ensure maximum winter values were at the ice saturating RH. Only one of the two corrected data sets is shown for clarity.

### 3.4 Wind speed

Data are from a cup anemometer at 2.3 m height. In winter the anemometer frosted up during some stable weather periods and was blocked about 2-3 weeks each year. Gaps were filled with data from Young anemometers at SILA, CAMP or SALIX, adjusted using correlations. Often, two or three of the anemometers were blocked simultaneously. One gap could not be filled in November 2018 because all anemometers were blocked. Since these blocking episodes all happened under low windspeed (<2 m/s, usually much less), that 2-week gap was filled with similar low value data from another period. Since the threshold value for cup anemometers is higher than for Young anemometers, when the cup value was 0 for extended periods, we used the value provided by the Young from CAMP. Each time, we checked that the CAMP values were quite low, and in any case <0.4 m/s. The wind speed time series is shown in Figure 2. A flag indicates gap-filled values (0=data; 1=gap-filled). 842 out of 52202 values were gap-filled (1.6%).

### 3.5 Atmospheric pressure

We did not measure atmospheric pressure. We relied on measurement performed by Environment and Climate Change Canada (ECCC, https://climate.weather.gc.ca) who operate a station at Pond Inlet (N:72.6951; W:77.9600), 60 m a.s.l., 84.1 km to the SE of our site and another station at Cape Liverpool (N:73.6681; W:78.2942) 2 m a.s.l., 79.5 km to the NE of our site. We present the average of both values, bearing in mind that the altitude of our site is probably close to 20 m. The atmospheric pressure time series is shown in Figure 2.

## 3.6 Long wave downwelling radiation

The pyrgeometer provides a raw signal, which is the difference between LW↓ and its own LW emission. The raw data were corrected for the drift in sensor sensitivity, which decreased from 6.57 µV W$^{-1}$ m$^2$ to 6.055 µV W$^{-1}$ m$^2$, an 8.5% decrease, assuming a constant drift over the 2193 days of use. LW↓ is then obtained by removing the instrument LW emission, and this is done using the temperature value provided by the CNR4 temperature sensor. Figure 4 shows that the raw signal remained close to zero over extended periods. This indicates similar temperatures for the sensor and the source (upper atmosphere or clouds), which is only possible if the source is close to the sensor. This is explained by the presence of frost on the pyrgeometer window. The 5-minute hourly heating by the CNF4 was therefore not sufficient to prevent frost build-up. Figure 4 however shows some periods in winter with no frost, e.g. January 2016. The CNR4 temperature sensor data were used to perform the the LW↓ correction and obtain the true LW↓ data over those periods. These were used to determine the correlation between ERA5 reanalyses (Hersbach et al., 2020) and our values. ERA5 values were noticeably lower than CNR4 ones. For winter (i.e. here the 21 October to 14 April period, which includes all frost-affected events) non-frost-affected values the correlation is:

$$LW{\downarrow}_{CNR4} = LW{\downarrow}_{ERA5} \text{ x } 0.7900 + 57.814 \text{ W m}^{-2} \qquad (3)$$

This is based on 10850 values and $R^2$=0.50. Invalid data were replaced with ERA5 values modified with equation (3). During the first year the CNR4 temperature sensor was not functioning, preventing the temperature correction of the raw signal. The correlation between all valid CNR4 values (including summer) and ERA5 values was:

$$LW{\downarrow}_{CNR4} = LW{\downarrow}_{ERA5} \text{ x } 0.9233 + 41.777 \text{ W m}^{-2} \qquad (4)$$

Equation (4), with $R^2$=0.79, was used to obtain a time series of LW↓ for the first year of this study. It could be argued that for winter 2013-2014 eq. (3) should be used. However, for the low LW↓ values of that period, the difference between eqs. (3) and (4) is small, between 0 and 10 W m$^{-2}$. Furthermore, a sudden change in equation may create an undesirable discontinuity. A flag indicates gap-filled values (0=CNR4 data; 1=modified ERA5 data). Overall, 21126 out of 52202 LW↓ values were gap-filled (40%).

The CNR4 and ERA5 LW↓ values are shown in the same graph in Figure 5 for comparison. Over the whole period investigated LW↓$_{CNR4}$ values are higher than LW↓$_{ERA5}$ values by an average value of 24.5 W m$^{-2}$. This large difference is discussed and the validity of our measurements confirmed when we present LW↑ values in section 4.2.

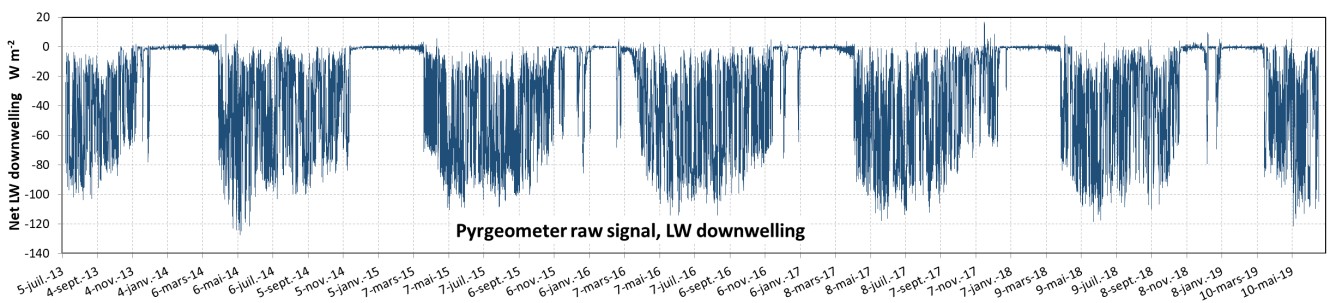

**Figure 4. Raw longwave downwelling radiation measured by the CNR4. The near-zero values in winter over extended periods indicate the presence of frost on the pyrgeometer window. The data for those periods are thus invalid.**

### 3.7 Short wave downwelling radiation

The calibration constant for the upward-looking pyranometer of the CNR4 drifted from 15.37 to 15.12 $\mu V\ W^{-1}\ m^2$, a 1.65% change. The data were adjusted for this drift. The SW↓ radiation showed a -2 W m$^{-2}$ offset and a value of 2 W m$^{-2}$ was added to all data. As for the pyrgeometer, frost built up on the upper pyranometer window. However, since there is no SW↓ to measure at 73°N most of the winter, data are overall less affected. In late March and early April, and to a lesser extent in late October, some albedo values were anomalously low, around 0.4. The corresponding SW↓ were anomalously high (Figure S5), much higher than ERA5 values, and this happened during periods when pyrgeometer data indicated the presence of frost, and when the sky was clear. Our CNR4 is over essentially flat terrain, so slope effects (Picard et al., 2020) can be ruled out. We propose that frost sublimated on the south side of the hemispherical pyranometer window, allowing direct radiation to reach the pyranometer, while it remained on its north side, scattering extra radiation into the pyranometer. The pyrgeometer window is flat, so frost there is less likely to be sublimated by radiation. Detailed data analysis showed that this process was more intense for solar zenith angles between 71 and 81°. While we cannot explain this in full detail, it is consistent with the idea that there exists an optimal geometry to maximize the scattering effect we propose. For those SW↓ data, we replaced them with corrected ERA5 data. For non-frosted conditions, ERA5 and CNR4 values are extremely well correlated:

$$SW{\downarrow}_{CNR4} = SW{\downarrow}_{ERA5}\ x\ 0.9956 - 3.7033\ W\ m^{-2} \qquad (5)$$

The RMSD is 19.51 W m$^{-2}$. For frost-affected periods, we therefore used equation (5) to obtain SW↓ values. However, since some ERA5 values were probably underestimated, this resulted in some albedo values > 1, which is not consistent with a sound radiation budget. Some data users may decide to modify some of the ERA5-derived SW↓ values presented here to ensure a reasonable albedo value, probably around 0.8. However, we left the ERA5-derived SW↓ values unchanged, leaving decisions regarding modification to the data users. A flag specifies when modified ERA5 values were used (0=CNR4 data; 1=modified

ERA5 data). Overall, 2550 out of 52202 SW↓ values were gap-filled (4.9%). Finally, instrument noise yielded non-zero data even during the polar night. We set SW↓ to 0 when the ERA5 reanalysis value was 0.

A slight tilt due to ground freezing and thawing, always less than 1.5° and usually less than 1° was observed most years on the CNR4 during maintenance. Given the generally high solar zenith angle, this may significantly affect SW↓ radiation measurement under clear-sky conditions. However, given the excellent correlation with ERA5, we conclude that these effect were probably not important. We are also providing ERA5 data for comparison but see no objective reason not to recommend our CNR4 data over ERA5 data. The SW radiation time series, both from CNR4 and ERA5, are shown in Figure 5.

### 3.8 Precipitation

There is a Geonor 200 precipitation gauge with a single alter shield at the CAMP site, 1.7 km from our TUNDRA site, but most of the time, it did not function properly. We therefore relied mostly on data from the ECCC Geonor gauges at Pond Inlet, 84.1 km to the SE, and Cape Liverpool, 79.5 km to the NE (Figure 1). Geonor gauges measure the weight of a glycol bath into which precipitation falls. There is noise in the data, for example due to wind, which produces small positive and negative precipitation events that need to be corrected and filtered. ECCC does not detail their procedure. The negative signals from their gauges appear to have been compensated (e.g. by reducing the positive signal from the subsequent positive event). There are numerous small (<0.2 mm in their hourly data) isolated positive event and we wondered whether those might just be noise. We examined isolated events that had no other precipitation 10h before and after them. By comparing these events to our time lapse photos and by considering the observer's remarks at Pond Inlet, we concluded that most of these isolated events were real, even though about 30% of them may be noise, because e.g. our cameras revealed blue sky conditions. We estimate that errors due to such noise amount to less than 4 mm per year. Precipitation needs to be corrected for undercatch under windy conditions and we used the equations of (Kochendorfer et al., 2017) for rain and snow. The threshold for rain/snow was set at +0.5°C. Examination of the air temperature when observations at the Pond Inlet airport indicated that precipitation was snow (on the ECCC web site) reveal that this threshold is sensible. We therefore used the gauge data from both ECCC sites, determined the phase at each site from the temperature there, also given by ECCC, and corrected the amount of precipitation using the local wind speed given by ECCC. To obtain precipitation at our site, we averaged both ECCC values and determined the phase at our site from our temperature measurement.

There were a few data gaps in the ECCC data sets. In that case we just used data from one of the two sites. There was a gap at Cape Liverpool from 30 November 2017 to 15 April 2018 (115 days) and two gaps at Pond Inlet from 12 to 29 April 2016 (18 days) and 13 February to 11 April 2017 (58 days). There was also a 24-day period from 8 September to 1 October 2017 when our gauge at the CAMP site functioned well and measured a greater amount pf precipitation than the ECCC gauges. During that period, we therefore used the CAMP data. Figure 5 shows hourly precipitation time series, separated as rain or snow. Overall, considering errors in undercatch correction, the distance between the instruments and our site, and the instruments' noise, we estimate the error on the precipitation data provided to be around 20%.

We also provide cumulated seasonal precipitation data for periods when there was snow on the ground and periods when the ground was snow-free. Snow onset is the first day when there is a continuous and permanent snow cover. Often, the first snow fall melted partially or completely, so that there is some arbitrary character in determining the snow onset date. For example, on 7 September 2017 a significant snowfall resulted in complete snow cover. That snow had mostly melted when an important snow fall that lasted the whole season happened on 17 September evening, so that we retain September 17 as the snow onset date. A picture on 17 September (Figure S6) shows what was left of the 7 September snowfall to illustrate our choice. Meltout date is when the winter snow cover has almost completely disappeared. Large snow drifts melt later. A picture in Figure S6 shows these remaining drifts on 8 June 2019, when we consider the snow had melted out. Occasional late spring snowfalls that occur after meltout were added to the summer precipitation. Snow onset and meltout dates were determined from snow gauges (present at TUNDRA and CAMP) and, when available, time lapse photographs. For 2013 and 2014, before the deployment of several time-lapse cameras in the valley, we also used satellite images to determine snow dates, as detailed in (Domine et al., 2018b). Table 2 reports the snow onset and meltout dates that we used. Cumulated seasonal precipitation time series are shown in Figure 5. Note that winter 2013-2014 was an exceptionally low-snow year.

**Table 2. Snow onset and meltout dates at the TUNDRA site, used to determined cumulated seasonal precipitation.**

| Snow year | Snow onset | Meltout |
|-----------|------------|---------|
| 2013-2014 | 11 October 2013 | 7 June 2014 |
| 2014-2015 | 12 September 2014 | 13 June 2015 |
| 2015-2016 | 1 October 2015 | 15 June 2016 |
| 2016-2017 | 3 October 2016 | 18 June 2017 |
| 2017-2018 | 17 September 2017 | 15 June 2018 |
| 2018-2019 | 8 October 2018 | 7 June 2019 |

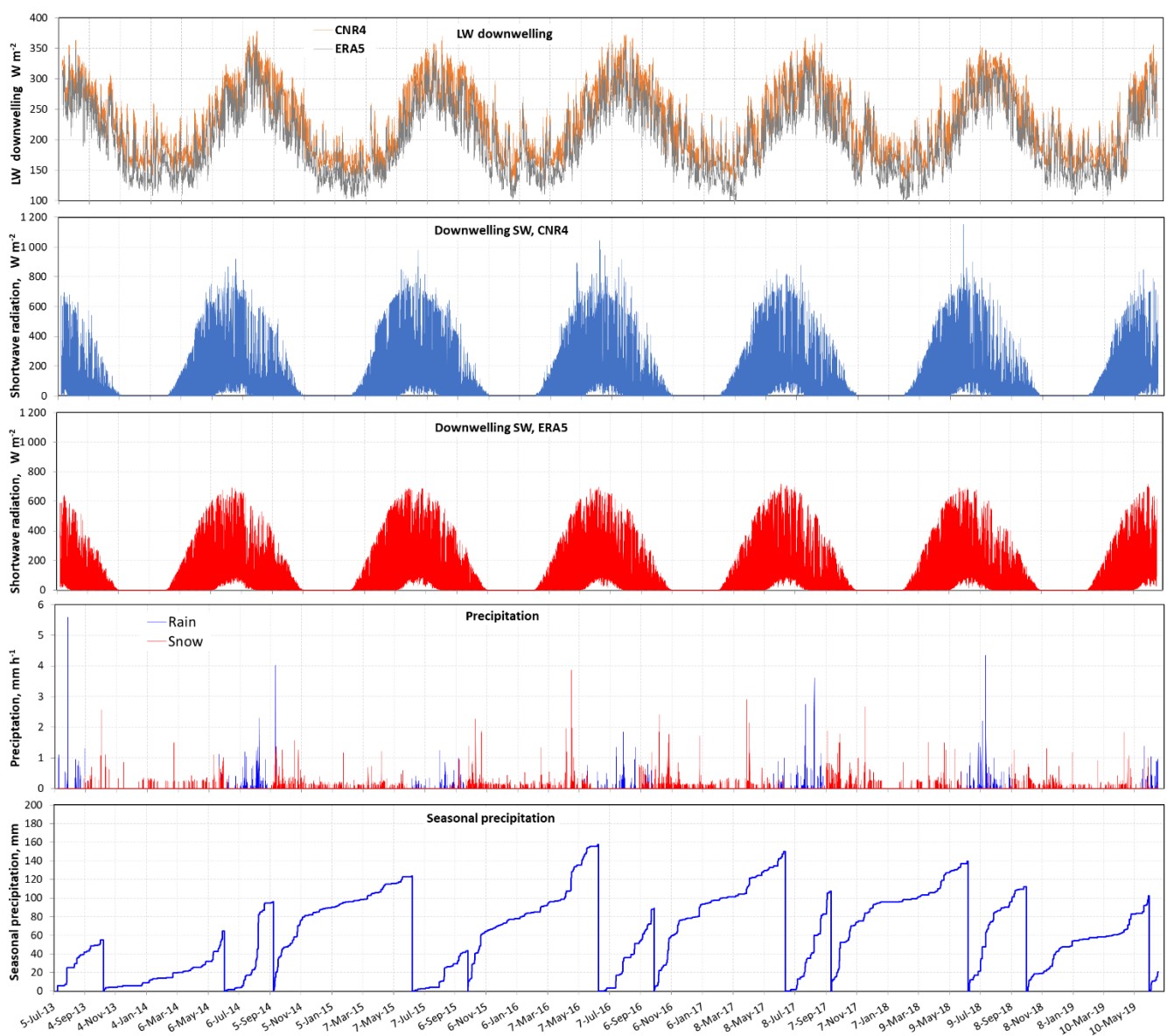

345 **Figure 5. Time series of downwelling long-wave radiation from our CNR4 pyrgeometer and from ERA5 reanalyses, downwelling short-wave radiation, both from our CNR4 pyranometer and from ERA5 reanalyses, hourly precipitation and cumulated seasonal precipitation (snow/snow-free periods).**

## 4 Validating data quality check and correction

Data for validation consist of monitoring data and snow pit measurements and observations every year in May except in 2016.

## 4.1 Short-wave upwelling radiation and albedo.

CNR4 values for SW↑ radiation were corrected for sensitivity drift, similarly to SW↓ radiation. The sensitivity changed from 15.74 to 15.39 μV W$^{-1}$ m$^2$ between 2013 and 2019, a 2.27% change. The downward looking pyranometer does not frost-up. Values were set to zero when ERA5 SW↓ values were zero. The error due to the tilt of the CNR4 discussed in the case of the downwelling radiation is probably negligible here, since radiation is diffuse. SW↑ values are affected by the presence of the tripod and of the solar panel it carries. Ideally, corrections can be performed, as done e.g. by (Wright et al., 2014). Values given here are uncorrected for the presence of the tripod and solar panel but we show in Figure S7 the geometry of the system so that the calculations could be performed. However, (Wright et al., 2014) had a geometry less favorable than ours and their correction was on the order of 1% so we do not expect this correction to be essential. Values thus obtained are reported in Figure 6. The high SW↑ value of 843 W m$^{-2}$ on 3 June 2018 at noon is most likely real since it corresponds to a high SW↓ value of 1151 W m$^{-2}$ at the same time (Figure 5). Other high values also correspond in the SW↑ and SW↓ data. These high values can be caused by thin clouds over snow which cause multiple reflections and amplify radiation. Partial cloud cover can also lead to significant radiation amplification. These effects have been predicted long ago (Nack and Green, 1974) and have been evidenced by studies focusing on UV radiation, with amplification sometimes exceeding a factor of 2 (McKenzie et al., 1998;Weihs et al., 2000;Lee et al., 2015) but the processes are similar for visible wavelengths and very likely explain these high values. ERA5 values do not seem to account for these processes and this is one reason why we recommend the use of our SW↓ data over ERA5.

Albedo obtained from the SW↑/SW↓ ratio is shown in the same panel as SW↑ in Figure 6. Values very different from 0.8 during snow-covered periods are due to the use of modified SW↓ ERA5 values, while SW↑ are all from the CNR4. A flag in the data file indicates which albedo data used ERA5-derived SW↓ values (0=CNR4; 1=SW↓ modified ERA5).

## 4.2 Long-wave upwelling radiation and surface temperature

Both the downward-looking pyrgeometer and the IR120 surface temperature sensor provide information on LW↑ radiation. The CNR4 LW↑ sensor sensitivity changed from 6.38 to 5.95 μV W$^{-1}$ m$^2$ between 2013 and 2019, a 7.23% change and the signals were corrected accordingly. As for the LW↓, no data are available the first year. The IR120 provides surface temperature $T_s$ but was not recalibrated. $T_s$ and LW↑ are linked by the Stefan-Boltzmann equation so that IR120 data may be used to fill the first year of missing data from the CNR4 LW↑. Both data sets are quite similar. The RMSD between the CNR4 LW↑ and the IR120 LW↑calculated from $T_s$ using a surface emissivity ε=1 for the 2014-2019 period is 7.51 W m$^{-2}$. The RMSD can be reduced if different ε values are used during snow-covered and snow-free periods. For the 2014-2015 winter, RMSD=6.12 W m$^{-2}$ is obtained for ε =1.027. For the 2015 summer, RMSD=10.54 W m$^{-2}$ is obtained for ε =0.991. Obviously ε cannot be >1, and the 1.027 value only indicates a systematic shift between both sensors. This is not surprising as the wavelength range sensed by the IR120 is narrower (Table 1) and small errors in the calibrations are inevitable. By comparing

CNR4 drift-corrected data and IR120 uncorrected data over 5 years, we did not however note any detectable drift in the IR120 sensitivity. For the first year, the CNR4 LW↑ data gap was filled with the IR120 data, with the optimal emissivities found above for the snow-covered and snow free periods. A flag in the data file indicates IR120-filled data. CNR4 LW↑ time series and surface temperature time series from IR120 are plotted in Figure 6. IR120 data have not been modified.

385    We compared our CNR4 LW↑ data with ERA5. ERA5 LW↑ was on average 16.7 W m$^{-2}$ lower, showing that ERA5 underestimates the temperature of the surface. The simplest explanation is that ERA5 LW↓ is underestimated, and this reduces surface warming. This confirms that our LW↓ data, which are on average 24.5 W m$^{-2}$ higher than ERA5, are probably correct and we recommend their use over ERA5 for our site.

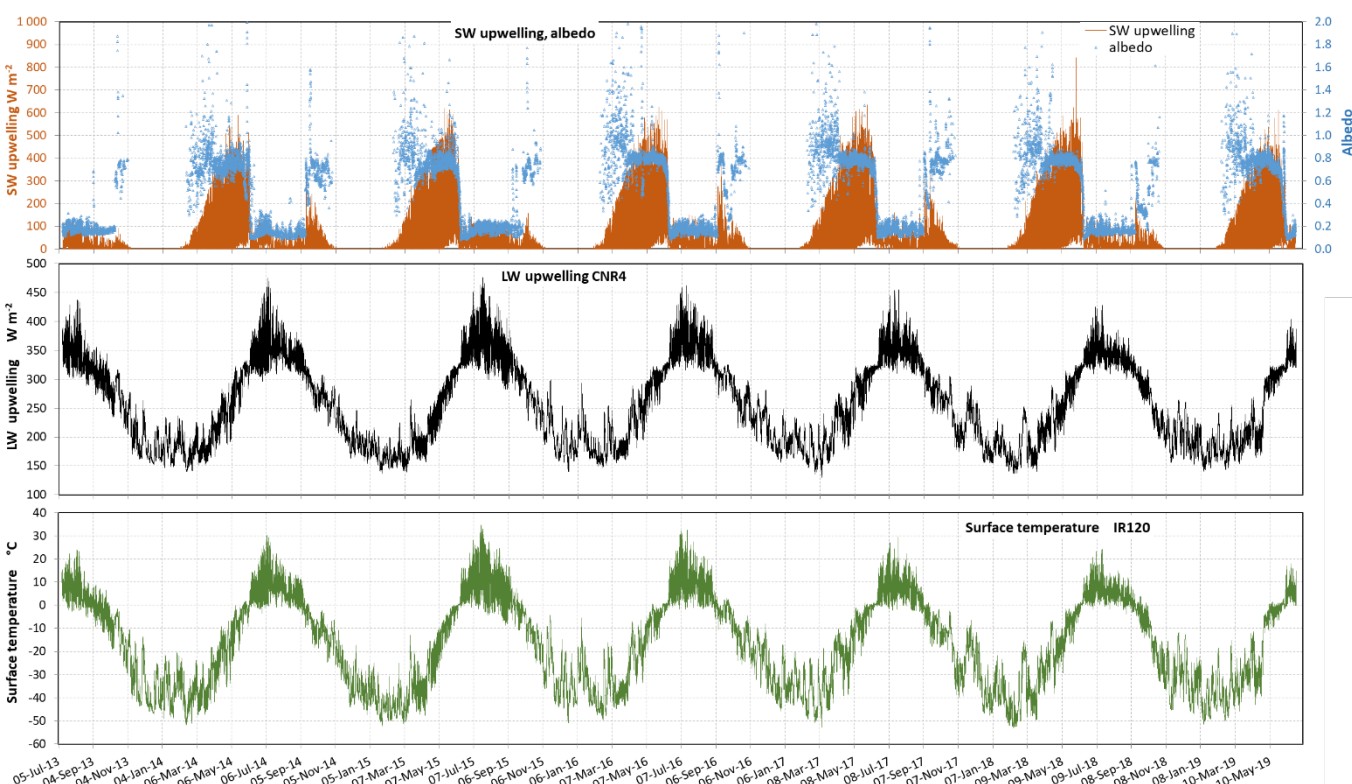

390    **Figure 6. Time series of SW↑ radiation from the CNR4 radiometer, albedo, LW↑ radiation from the CNR4 radiometer and surface temperature from the IR 120 infrared sensor. The albedo values are shown with a reversed scale to minimize overlap with the SW↑ plot. Most albedo values over snow-covered surfaces that are very different from 0.8 are due to the use of ERA5 SW↓ values.**

### 4.3 Snow depth

395    Continuous snow depth data from the TUNDRA snow gauge are shown in Figure 7. To facilitate reading, snow-free periods were assigned a zero snow depth value. However, snow depth is highly spatially variable because of the relief at the 10 to 20 m scale in the ice-wedge polygon terrain. Therefore, additional manual snow depth measurements were taken in May 2014,

2015, 2017, 2018 and 2019 at several hundred random spots around the tundra site. The means and standard deviations are shown in Table 3. Snow depth measurements were also done in numerous spots in the whole valley. This confirmed that spring 2018 was indeed the snowiest year we experienced, and spring 2014 by far the lowest snow depth everywhere in the valley. The snow depth data of Table 3 is therefore representative of the climatology at least at the 20 km scale.

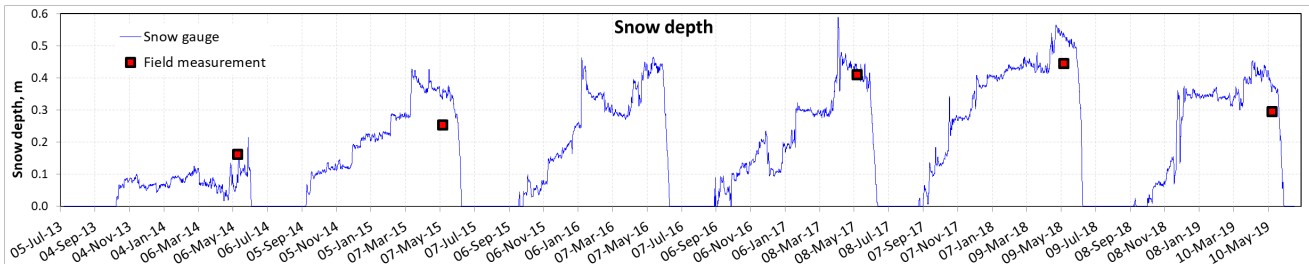

**Figure 7. Time series of snow depth monitored by an automatic snow gauge. The averages of over 100 spot measurements in mid-May around the TUNDRA site are also shown for five of the 6 years.**

**Table 3. Mean values and standard deviation of snow depths measured around the TUNDRA site in May.**

| Date | Mean depth | Standard deviation |
| --- | --- | --- |
| 14 May 2014 | 16.2 cm | 13.7 cm |
| 12 May 2015 | 25.3 cm | 13.1 cm |
| May 2016 | No data | No data |
| 13 May 2017 | 41.0 cm | 10.9 cm |
| 14 May 2018 | 44.5 cm | 13.4 cm |
| 17 May 2019 | 29.5 cm | 13.8 cm |

### 4.4 Snow temperature

Snow temperatures were measured with Pt100 thermistors installed in July 2014. For the 2013-2014 season, we provide temperature given by the TP08 heated needle probe, which produced one value every other day at 5:00 local summer time, when a thermal conductivity measurement was performed. In 2014-2015, there were only 2 thermistors, at 2 and 17 cm heights. In July 2015, thermistors were added at 7, 27 and 37 cm. In July 2018, all 5 thermistors were lowered by 2 cm to 0, 5, 15, 25 and 35 cm. All data >0°C were deleted. Data when no snow was present on the ground have also been deleted, based on snow height data or time lapse images. However, the snow gauge is about 6 m away from the thermistor post and only the top of the post is in the field of view of the camera. Another criterion for the presence of snow is the temperature gradient in the set of

sensors. When snow is present, the lowest sensor is expected to be warmer, at least until spring warm up, when the temperature gradient reverses. However, all these criteria are not 100% certain, and we may have included some data in the absence of snow. Data from upper sensors not covered by snow have not been deleted. Snow temperature data are shown in Figure 8.

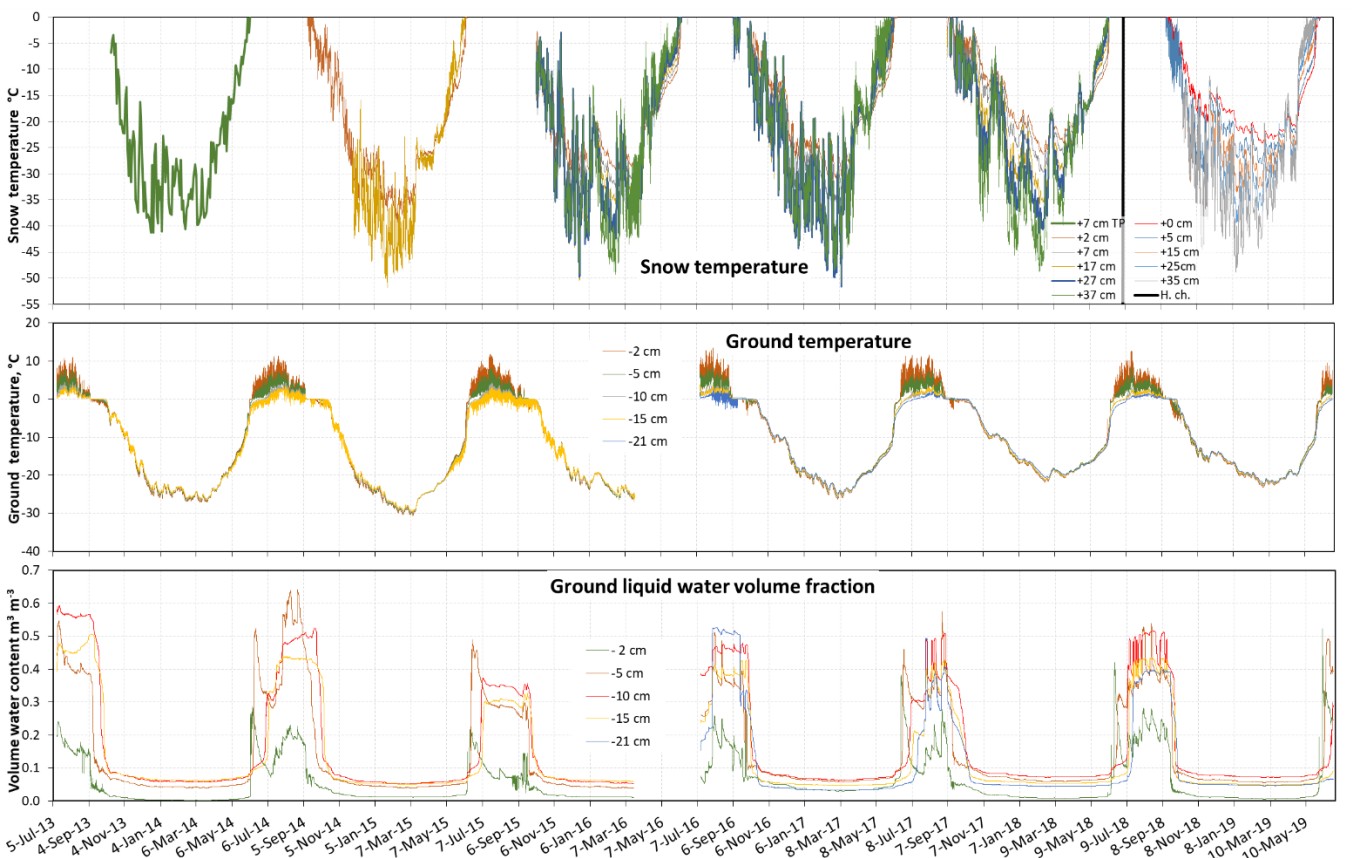

**Figure 8. Time series of snow temperature from Pt100 thermistors and ground temperature and liquid water volume fraction from 5TM probes. In 2013-2014, snow temperature data was limited to 7 cm height (low snow height that year) with a reading from the TP08 probe every other day at 5:00. The heights of the snow temperature sensors were lowered by 2 cm in July 2018. The vertical black line in July 2018 shows the date of this height change (H. ch. in legend box). There are negative spikes due to instrumental noise on the soil temperature at 15 cm depth data until 2016 and on the 21 cm depth data in summer 2016.**

### 4.5 Ground temperature and liquid water volume content

These variables were measured using 5TM sensors from Decagon placed within 1 m of the TP08 post. The deepest sensor was placed at the base of the summer thawed layer. Decagon documentation specifies "the 5TM determines volumetric water content (VWC) by measuring the dielectric constant of the media using capacitance/frequency domain technology. The sensor uses a 70 MHz frequency, which minimizes textural and salinity effects, making the 5TM accurate in most soils. The 5TM

measures temperature using an onboard thermistor." Regarding temperature, offsets of up to 0.5°C, constant over time, were noticed during soil freezing. All temperatures were corrected so that T=0°C during the zero-curtain periods. Regarding VWC, the calibration provided by Decagon was used. For mineral soils, a 2% accuracy is claimed by the manufacturer. For other soils, 3% is claimed. This lower accuracy probably applies to the top two sensors at 2 and 5 cm depth, where the soil has a significant organics content. Due to battery failure, there is a data gap between 20 March and 12 July 2016. Before March 2016, the temperature sensor at 15 cm depth showed very frequent spikes in summer that gave readings lowered by 1.5 to 3°C, which is why the plots appear noisy. The same applies to the 21 cm sensor in August 2016. The causes are unknown. Data are shown in Figure 8.

## 4.6 Snow and soil thermal conductivity

Measurement methods using the TP08 heated needle probe are detailed in (Domine et al., 2015). Data from the first 3 winters have already been reported in (Domine et al., 2016b) and (Domine et al., 2018a). Figure 9 shows measurements for all 6 years at 3 heights. In 2013-2014, only the 7 cm needle was covered. In July 2014 the sensors were lowered to 2, 12 and 22 cm.

Soil thermal conductivity values only show significant variations between the thawed and frozen state, as frequently observed in soils (Smerdon and Mendoza, 2010). Thawed and frozen values are around 0.75 and 1.8 W m$^{-1}$ K$^{-1}$ respectively, with little variations between years. Values may vary with water or ice content but this was not investigated here. In the frozen state, many heating curves were of insufficient quality because of the limited heating and those data were discarded (see (Domine et al., 2015) for details), hence the missing data points.

Snow thermal conductivity is a valuable proxy for snow type. Soft depth hoar always has a low value and for example the very low thermal conductivity value at 2 cm height in 2014-2015 (mostly <0.035 W m$^{-1}$ K$^{-1}$) is indicative of the presence of very soft depth hoar, as observed in May 2015 during the field campaign. On the contrary, the high values in 2015-2016 (0.2 to 0.35 W m$^{-1}$ K$^{-1}$) indicate that depth hoar was probably indurated, due either to rain-on-snow (ROS) that formed a hard refrozen layer or to high winds during precipitation that formed a hard wind slab. On October 1$^{st}$ 2015 ROS took place just after snow onset and on 14-15 October a 36-hour storm with wind speeds exceeding 10 m s$^{-1}$ and precipitation in excess of 10 mm took place, so that either of both options is possible. We could not get to Bylot Island in spring 2016 for snow observations. However, snow pit observations near Pond Inlet on 15 May 2016 indeed revealed the presence of a 10 to 15 cm basal layer of indurated depth hoar.

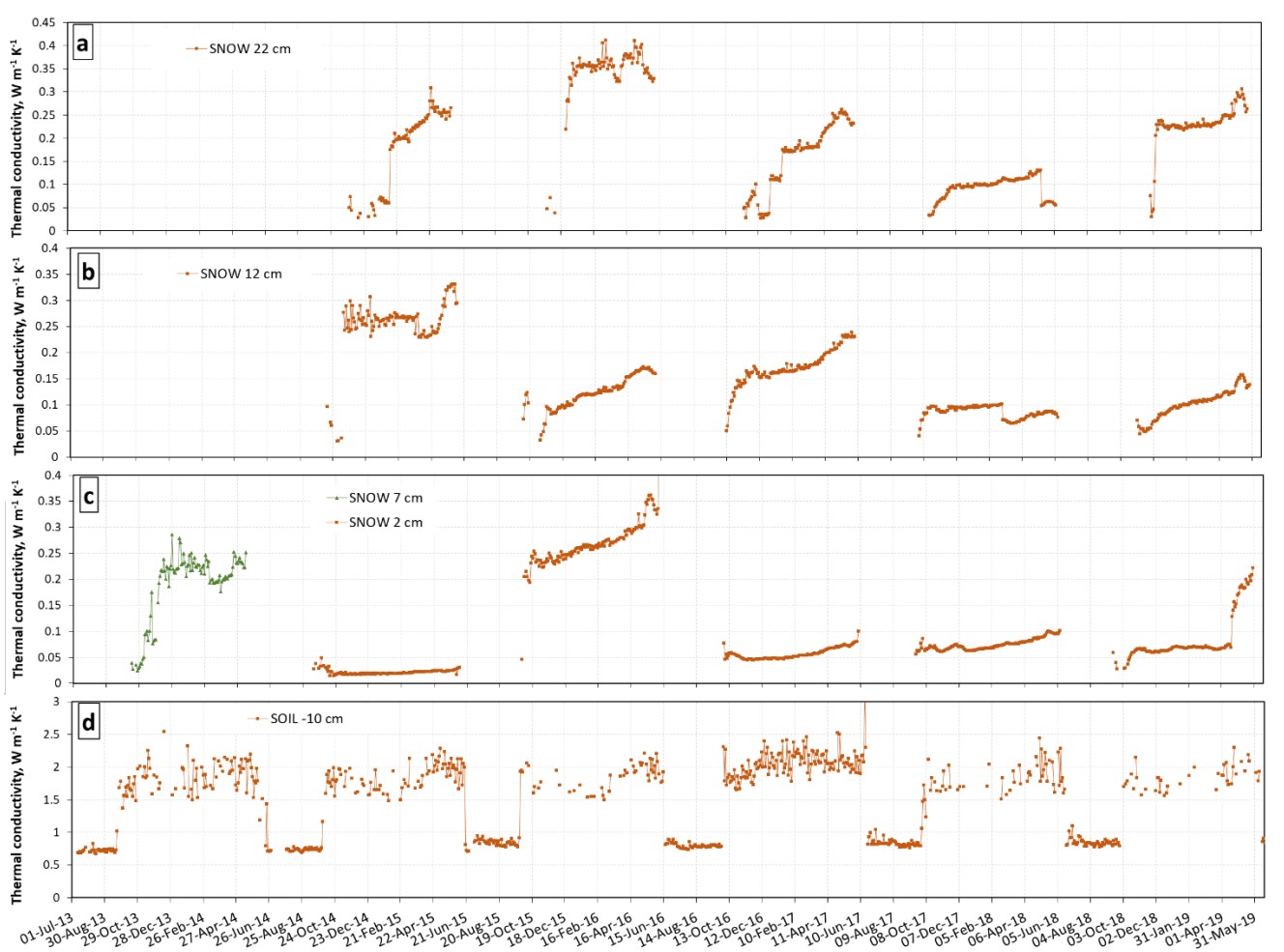

**Figure 9.** Time series of snow thermal conductivities at heights of (a) 22 cm; (b) 12 cm; (c) 7 cm for the first winter and 2 cm for subsequent winters; and (d) soil thermal conductivity at 10 cm depth. In 2013-2014 there was insufficient snow to cover the top two TP08 probes, which were at 17 and 27 cm heights.

It has been reported that the heated needle probe method produced a negative artifact in the measurement of snow thermal conductivity (Riche and Schneebeli, 2013). This is currently under investigation and a correction algorithm will be proposed by Fourteau et al. shortly. Briefly, the amount of correction decreases with increasing snow density and a multiplicative factor of about 1.1 for dense wind slabs and 1.5 for soft depth hoar must be applied. Data presented are uncorrected. Note that here the depth hoar thermal conductivity value at 2 cm in 2014-2015 has values around 0.02 W m$^{-1}$ K$^{-1}$, lower than air, and after correction these values will be around 0.03, more plausible for very light and uncohesive depth hoar.

### 4.7 Field observations of snow

Snow density and SSA profiles cannot today be monitored automatically. Instead, vertical profiles of these variables were measured at the TUNDRA site in mid-May 2014, 2015, 2017, 2018 and 2019 during field expeditions. The snow pits were dug in the actual polygon of the station, within 3 m of the snow thermal conductivity post. Data are shown in Figure 10. We stress here that these profiles are highly variable in space because of wind erosion and redeposition, which results in heterogeneous and often discontinuous snow layers. Attempting to reproduce the details of these profiles using 1-D model
simulations is therefore not very meaningful. To illustrate the spatial variability of these variables we report in Figures S8 and S9 additional profiles measured in the valley, in the absence of erect vegetation, i.e. in places where there is no *Salix richardsonii*, as these shrubs significantly affect snow properties (Domine et al., 2016a). The coordinates and dates of these additional profiles are reported in Table S1.

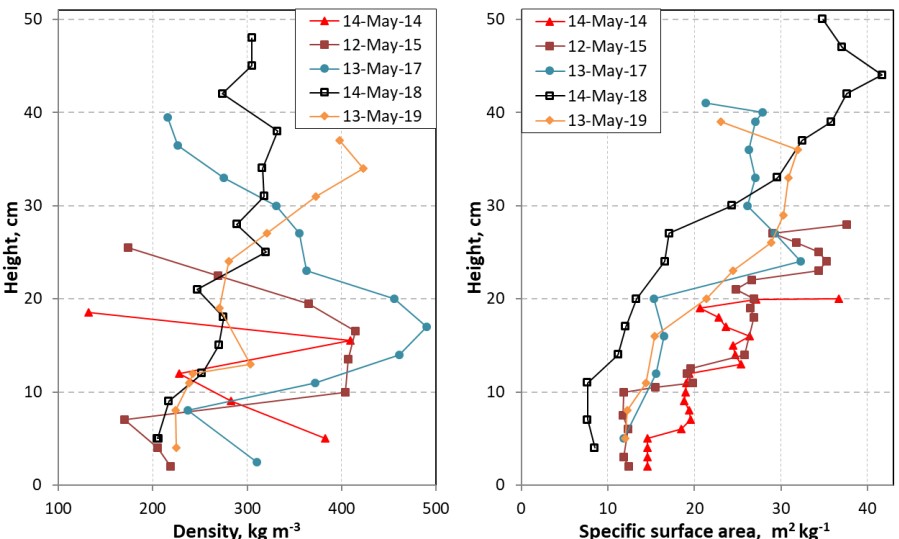

**Figure 10. Vertical profiles of snow density and SSA measured within the TUNDRA polygon in mid-May of 5 years.**

### 5 Conclusion

A 6-year time series of driving data for a high Arctic herb tundra site is presented. A unique set of validation data is provided which includes times series of snow and soil thermal conductivity. Vertical profiles of snow density and specific surface area in mid-May are also provided for all years except 2016. One important objective of these data is to assist in the improvement
and validation of snow physics models, which today have great difficulties in simulating high Arctic snowpack properties. We plan to update the data sets on the Nordicana D repository by adding extra years of data whenever possible. The COVID 19 pandemic prevented us from accessing the site in spring and summer 2020 and spring 2021 but we will do our best to maintain our effort in subsequent years.

## Acknowledgements

This work was funded by the French Polar Institute (IPEV), the Natural Sciences and Engineering Research Council of Canada (Discovery Grant and Northern Research supplement programs), the BNP-Paribas foundation (APT project) and the European Commission (INTAROS project). Logistical support was provided by the Polar Continental Shelf Program and by Sirmilik National Park. We are grateful to Gilles Gauthier and Marie-Christine Cadieux for their decades-long efforts to build and maintain the research base of the Centre d'Etudes Nordiques at Bylot Island. Assistance with field work by Mathieu Barrère,

Mikael Gagnon and Marianne Valcourt is gratefully acknowledged. Christophe Kinnard kindly pointed out an error in the Campbell program for the CNR4. This allowed us to make the best use of available radiation data. We acknowledge helpful discussions with Ghislain Picard and Marie Dumont regarding LW and SW radiation, and comments on the manuscript by Warwick Vincent. We thank Richard Essery and Cécile Ménard (U. Edinburgh) for expressing interest in this work and encouraging its completion. Compute Canada assisted with data storage and handling. This work is a contribution to the IASC

project Terrestrial Multidisciplinary distributed Observatories for the Study of Arctic Connections (T-MOSAiC).

## Data availability

The driving and validating data, including snowpit data, are available on the Nordicana D repository, https://doi.org/10.5885/45693CE-02685A5200DD4C38 (Domine et al., 2021).

## Authors contributions

FD designed research and obtained funding. DS and FD deployed and maintained instruments. FD and GL analyzed data and prepared the data files. MP, FD and MBB performed the field work. FD wrote the paper with inputs from GL and comments from MBB, DS and MP.

## Competing interests

The authors declare that they have no conflict of interest.

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
