# Peer review of "Meteorological, snow and soil data (2013-2019) from a herb tundra permafrost site at Bylot Island, Canadian high Arctic, for driving and testing snow and land surface models"

_Earth System Science Data, 2021_

## Author Comment (AC1)

**Response to Reviewer 1.**

*Our responses are in blue italics, embedded into the reviewer's comments.*

The authors introduce 6 years of meteorological, soil and, snow data collected at a tundra site in the Canadian high Arctic. The observed polygonal tundra environment located on Bylot Island, Nunavut is described as well-drained, void of standing vegetation, and typical of permafrost landscape. A thorough review of instrumentation, datasets, and validation at the site are presented prior to an assessment of data quality and correction of the temporal data where necessary. The authors highlight a lack of data available to drive snow physics models in herb (graminoid) tundra environments as motivation. Thin Arctic snowpack common to this area, and greater Arctic regions, consist of contrasting high density winds slab and lower-density depth hoar elements. The predominate drivers of this composition, wind compaction and upward flux of water vapour, are clearly identified by the authors as a gap in the current generation snow physics models requiring new datasets to evaluate against and improve from.

The described work is a valuable contribution and should be published in ESSD. Public release of this dataset and its associated validation are a service to the snow, climate and meteorological communities where significant effort has been required to initiate and sustain these remote measurements. I have noted instances below where I felt minor clarifications were needed on corrections applied but in general, I felt the dataset was well documented. Where internet data resources were accessed, for example the ECCC station data, date of access should be noted in the references in case of future revision. Useful lessons and anecdotes on sensor deployment, calibration and adaptation punctuate the paper and I felt it could be more impactful if these were summarized in a concise discussion near the end. A discrete guide on how to setup an Arctic site is clearly beyond scope, but some of the more significant choices, for example the use of ERA5 over the CNR4 data, warrant clear recommendations. I would hope such as summary would be a catalyst for development of similar sites, amplifying the efforts here.

*We thank Joshua King for his overall positive recommendation and carefully consider his minor recommendations below.*

Minor comments with line numbers provided:

Lines 47, 68,73 and others – Where the authors are part of the narrative should the citations not be in-line instead of in parenthesis?

*Yes indeed. This is how endnote works and it will be fixed in the final version or at type setting.*

Line 149: What does significant mean in the contact of the spatial variability described here? It would be helpful to place this in the context of another study or even a metric of the locally observed variability to constrain the use of the word significant.

*We will specify "As reported in Figure S3 for our very site, and as also observed at other spots in the valley, the thermal conductivity at a given soil level can vary by a factor up to 3 within a few meters"'*

Line 165 Superscript is missing for ^3

*Thank you. This will be fixed.*

Line 167: Add the total number of pits completed in the sentence or point towards a table to summarize.

*Pit data are detailed in the results section and are shown in Figure S7 and S8. We will state here that "Depending on the year, 2 to 7 snowpits were studied on herb tundra".*

Line 181: It is not clear if the correction applied was a constant or varied over time.

*Thanks for having us look into this. In fact, we did not use the CNR4 sensor, but the datalogger sensor. We will correct our text and mention that for the correction was a function of incident solar radiation. It is therefore not a constant over the year when it was used. This will be specified.*

Line 198: This internet resource needs to be cited with the data of access.

*We will mention that it was accessed on 5 February 2021.*

Line 213: The gas station equation does not have its variables introduced in text with units.

*Indeed, this will be added, we apologize for the omission.*

Line 271 The Pond Inlet airport station has an observer. Were the observer comments ever used to evaluate selected precipitation phase threshold at +0.05C?

*It is in fact 0.5°C. Thank for pointing this out. We did not use this information in our data set. However, we plotted the air temperature during the rain events and also during the snow events, as indicated by the Pond Inlet airport data, and +0.5°C really seems like the most sensible boundary.*

Line 279: The use of time lapses appears to be key here. Was this approach contrasted against an automated or empirical approach?

*In fact, the camera was installed in summer 2015 so that we used the automatic snow gauge and satellite data to determine snow cover dates for 2013 and 2014. The procedure has been detailed in (Domine et al., 2018). For the other years, since we had a camera, we did not seek satellite data. The snow gauge gave the same snow-in and meltout dates as the camera, within one day. We will mention our method for 2013 and 2014. We feel there is no need to compare gauge and camera result since we never relied on the snow gauge only.*

Line 307: It was previous stated that there was no small-scale relief at the tundra site in the context of hummock-tussock formations. Is this related to the wedge structures?

*Indeed, we stated line 125 that "There is no small-scale topography within the polygon (Figure 1) and in particular no hummock or tussock." Now in line 306 we state "However, snow depth is highly spatially variable because of the small-scale relief in the ice-wedge polygon terrain." We realize this may be confusing and we will specify that the scales are different. The hummock-tussock relief is at the 50 cm scale. The relief in polygonal terrain is indeed due to the ice wedges and is at the 10 to 20 m scale. Thanks for spotting this problem.*

Line 395: This reference appears to be incomplete. Please expand on what type of correction was applied.

*There is in fact no citation possible because the relevant work by Fourteau et al. has been submitted to J. Glaciol. and submitted papers cannot be cited. We explain lines 395-396 that "the amount of correction decreases with increasing snow density and is about 1.1 for dense wind slabs and 1.5 for soft depth hoar." Fourteau et al. propose density-based corrections. In any case, the fact that we cannot detail this now has no impact on our current paper because we state line 396 "Data presented are uncorrected." We believe that no change on this point is warranted until the paper by Fourteau et al. is accepted.*

*Finally, we wish to point out that we have solved the issue concerning our long-wave (LW) downwelling data, which we did not trust, so that we used ERA5 data instead. This was due to an error in the program supplied by Campbell scientific during 2012-*

*2013 to run the CNR4. That program applied the sensitivity coefficient of the downwelling SW sensor to all four sensors. This has been corrected. However, we also noticed that frost built up on both CNR4 upper sensors and that the 5 minutes of hourly heating were often not sufficient to remove the frost. We therefore built a hybrid downwelling LW data set, where periods impacted by frost were replaced by ERA5 data corrected using a correlation between valid CNR4 winter data and ERA5. In the revised version, we will therefore include a hybrid downwelling LW data set. The upwelling SW will also be corrected for the error in the Campbell program.*

**Reference**

Domine, F., Gauthier, G., Vionnet, V., Fauteux, D., Dumont, M., and Barrere, M.: Snow physical properties may be a significant determinant of lemming population dynamics in the high Arctic, Arctic Science, 4, 813-826, 2018.

---

## Author Comment (AC2)

**Response to Reviewer 2**

*Our responses are in blue italics, embedded into the reviewer's comments.*

**general comments**

The authors describe and provide an extensive set of in-situ meteorological, snow and soil data to force and evaluate snow schemes in a tundra environment. Such an extensive set of forcing and evaluation data, especially for snow, is unprecedented in the litterature in this type of environment. As the most complex snow models to date fail to represent tundra snow charateristics, despite their huge significance at the global scale and w/r to global warming, this paper and dataset are in my opinion an important contribution to snow science.

*Thank you for this encouraging appreciation.*

The site and the dataset are well described, the quality assessment of the data is thorough, and the corrections performed to the data are well explained and justified (see just some minor clarification needs in the specific comments). I recommand the paper for publication in ESSD after these minor comments have been addressed.

*Thank you for this positive evaluation.*

**specific comments**

L51-52: "The explanation proposed (Domine et al., 2019;Domine et al., 2016b) is that Crocus and SNOWPACK were designed primarily for avalanche forecasting in the Alps, i.e. for mid latitude warm thick snowpacks while the Arctic features cold thin snowpacks." It would seem more faithful to me to cite also the other applications of these models besides avalanche forecasting (something like " were designed for avalanche forecasting and process-studies/other applications in the Alps")

*Indeed, we agree. Even though the original main motivation for developing these models was avalanche, they have been used for a wide range of application, and in particular for land surface studies, and climate and hydrological issues (Brun et al., 2013; Barrere et al., 2017; López-Moreno et al., 2020). This will be mentioned. Thank you for pointing this out.*

- L58-59 : "This process is not simulated by Crocus or SNOWPACK, leading to erroneous outputs." I think the attempts to partially account for this process in both models (Touzeau et al., 2018 and Jafari et al., 2020), should be mentioned here. Touzeau, A., Landais, A., Morin, S., Arnaud, L., and Picard, G. (2018). Numerical

experiments on vapor diffusion in polar snow and firn and its impact on isotopes using the multi-layer energy balance model crocus in surfex v8.0. Geosci. Model Dev. 11, 2393–2418. doi: 10.5194/gmd-11-2393- 2018

Jafari M, Gouttevin I, Couttet M, Wever N, Michel A, Sharma V, Rossmann L, Maass N, Nicolaus M and Lehning M (2020) The Impact of Diffusive Water Vapor Transport on Snow Profiles in Deep and Shallow Snow Covers and on Sea Ice. Front. Earth Sci. 8:249. doi: 10.3389/feart.2020.00249

*This is true, both these papers attempt to include water vapor diffusion in Crocus and SNOWPACK. However, they do not model the water vapor transport process involved in Arctic snow, which is convection. Convection is at least an order of magnitude more efficient at transporting water vapor than diffusion, as already indicated decades ago by (Benson and Trabant, 1973) and confirmed by studies in Alaska (Johnson et al., 1987; Sturm and Benson, 1997) and at Bylot Island (Domine et al., 2016). (Touzeau et al., 2018) do a fine job at modeling diffusion in firn on ice sheets but "focus on the movement of water isotopes in the vapor in the porosity, in the absence of macroscopic air movement." (page 2394, 2nd column, 2nd paragraph). Their work is therefore not relevant to Artic snow and will not help Crocus performance in the Arctic. Their objective in any case was for isotopic exchange on ice sheets and not mass distribution in Arctic snow. The work of (Jafari et al., 2020) details water vapor diffusion in Arctic snow but to obtain a significant mass transfer these authors have to use water vapor diffusion coefficients in snow that are significantly greater than in free air. Latest work of the subject demonstrate that water vapor diffusion in snow is lower than in free air (Fourteau et al., 2021), although we admit this is a controversial topic. Furthermore, (Jafari et al., 2020) state "We acknowledge that vapor transport by diffusion may in some snow covers— such as in thin tundra snow—be small compared to convective transport, which will have to be addressed in future work.", in their abstract. In any case, we feel it would be misleading to suggest that Crocus or SNOWPACK are able to simulate water vapor transport in Arctic snow. To avoid any misunderstanding, we will stress that convection is the main process responsible for water vapor transport in Arctic snow and cite references on the subject.*

- L181 : "Based on several years of simultaneous temperature measurements of the HCS2-S3-XT and CNR4 sensors, we corrected the CNR4 sensor values. We found that there was no bias between the two temperature measurements and a RMSD=0.784°C" I am rejoining the comment from Referee #1 on this : Details on the correction (does it involve radiations ?) would be welcome as an help/ a reference for other people encountering similar problems at their sites

*As stated in our response to Reviewer 1, our initial text was in error and we used the data logger temperature sensor. Yes, radiation was a variable in the correction and this will be detailed.*

-L332 : "Figure 2 may also indicate a decrease in summer air temperature" The statement is too vague, please clarify.

*It is indeed too vague. Discussing a temperature trend based on just 6 years of data is in fact not very interesting. We will remove this statement.*

- L345: At several places like this one "However, all these criteria are not 100% certain, and there may be some data in the absence of snow." the honesty of the authors is much appreciated !

*Thank you. We feel it is important to state clearly the limits of our data set.*

- L358 : "The deepest sensor was placed just above the frozen soil layer" Maybe it would be more clear if you specify that this is the frozen layer at its summer position.

*We will replace this by "The deepest sensor was placed at the base of the summer thawed layer"*

- L 395 : "Briefly, the amount of correction decreases with increasing snow density and is about 1.1 for dense wind slabs and 1.5 for soft depth hoar" Is this a multiplicative factor ? Thanks in advance for clarifying this

*Yes, measured values have to be multiplied by a factor between 1.1 and 1.5, increasing with decreasing snow density. This will be reworded.*

- Dataset : As a suggestion for later updates of this dataset, a time-series flag for each meteorological variable indicating whether the data has been gapfilled, corrected, or is the raw measure, could be usefull and more precise/exhaustive than what is currently written in the paper (for wind speed for instance)

*We agree 100% and apologize for not having done that. We will certainly do that when the data set is incremented. In fact, we will start right now. For our new downwelling LW data, we will add a column to specify whether the data points are from the CNR4 or are ERA% modifies data.*

**References**

Barrere, M., Domine, F., Decharme, B., Morin, S., Vionnet, V., and Lafaysse, M.: Evaluating the performance of coupled snow–soil models in SURFEXv8 to simulate the permafrost thermal regime at a high Arctic site, Geoscientific Model Development, 10, 3461-3479, 2017.

Benson, C. S. and Trabant, D.: Field Measurements on the flux of water vapour through dry snow, in: Proceedings of the International Symposium on The Role of Snow and Ice in Hydrology, , 1973Banff, 6-13 September 1972 1973, 291-298, 1973.

Brun, E., Vionnet, V., Boone, A., Decharme, B., Peings, Y., Valette, R., Karbou, F., and Morin, S.: Simulation of northern Eurasian local snow depth, mass and density using a detailed snowpack model and meteorological reanalysis, J. Hydrometeorol., 14, 203-214, 2013.

Domine, F., Barrere, M., and Sarrazin, D.: Seasonal evolution of the effective thermal conductivity of the snow and the soil in high Arctic herb tundra at Bylot Island, Canada, The Cryosphere, 10, 2573-2588, 2016.

Fourteau, K., Domine, F., and Hagenmuller, P.: Macroscopic water vapor diffusion is not enhanced in snow, The Cryosphere, 15, 389-406, 2021.

Jafari, M., Gouttevin, I., Couttet, M., Wever, N., Michel, A., Sharma, V., Rossmann, L., Maass, N., Nicolaus, M., and Lehning, M.: The Impact of Diffusive Water Vapor Transport on Snow Profiles in Deep and Shallow Snow Covers and on Sea Ice, Frontiers in Earth Science, 8, 2020.

Johnson, J. B., Sturm, M., Perovich, D. K., and Benson, C.: Field observations of thermal convection in a subarctic snow cover. In: Avalanche Formation, Movement and Effects (Proceedings of a Symposium held at Davos, September 1986) IAHS pub. 162, Salm, B. and Gubler, H. (Eds.), IAHS, 1987.

López-Moreno, J. I., Soubeyroux, J. M., Gascoin, S., Alonso-Gonzalez, E., Durán-Gómez, N., Lafaysse, M., Vernay, M., Carmagnola, C., and Morin, S.: Long-term trends (1958–2017) in snow cover duration and depth in the Pyrenees, Int. J. Climatol., n/a, 2020.

Sturm, M. and Benson, C. S.: Vapor transport, grain growth and depth-hoar development in the subarctic snow, J. Glaciol., 43, 42-59, 1997.

Touzeau, A., Landais, A., Morin, S., Arnaud, L., and Picard, G.: Numerical experiments on vapor diffusion in polar snow and firn and its impact on isotopes using the multi-layer energy balance model Crocus in SURFEX v8.0, Geoscientific Model Development, 11, 2393-2418, 2018.

---

## Author Response (AR1)

Dear Editor,

We are submitting our revised version of ESSD-2021-54 by Domine et al. The Reviewers wrote in general highly favorable reviews and only suggested minor revisions. We however made significant improvements to the manuscript. Essentially, in the initial version, we were not able to produce field data for downwelling and upwelling long wave radiation (LW↓ and LW ↑, respectively. This was because we judged that our values were much too high and in particular about 50 W m$^{-2}$ higher than ERA5 reanalyses values.

In this letter, all manuscript lines mentioned are those in the tracked-changes version.

Our colleague Christophe Kinnard read our discussion paper and contacted us to warn us that the program supplied by Campbell Scientific to run the CNR4 radiometer with their CR1000-XT data logger had an error, in that the coefficients used to convert radiometer signals from SW↓ (downwelling shortwave radiation), SW ↑, LW↓ (downwelling longwave) and LW ↑ all used the coefficient of the SW↓ sensor. We thus corrected our data accordingly, and this produced more sensible values. However, we still had a problem with the upper sensor for LW↓ because in the polar night frost built up on its window over extended periods, giving wrong readings. We were able to correct for this using a correlation between our valid data obtained during frost-free periods and ERA5 reanalyses. We therefore are now able to provide what we believe is a valid data set for LW↓ data at our site. We feel this is very important because the energy budget of snow surfaces during the polar night is highly dependent on LW↓. Our values are on average 24.5W m$^{-2}$ higher than ERA5 values. However, by considering surface temperature and LW ↑ values from our site and from ERA5, we are able to conclude that our values are sensible and ERA5 values are underestimated. The fabrication of the LW↓ time series is detailed in section 3.6 (lines 245-282). The surface temperature and new LW ↑ data discussed in section section 4.2 (lines 396-415) confirms the validity of our approach. The relevant modified data files have been uploaded to version 1.1 of our data at https://doi.org/10.5885/45693CE-02685A5200DD4C38

Other modifications related to these new data series include reorganization and reordering of Figures, with in particular the addition of Figure 4, and the separate presentation of upwelling radiation and surface temperature in Figure 6.

Lastly, we also discussed the impact of frost that built up on the CNR4 windows on our SW↓ measurements in section 3.7, lines 287 to 322. Frost-affected values were replaced by corrected ERA5 reanalysis values. We also present a more detailed section 4.1 on SW ↑ and albedo, lines 376 to 395. To illustrate the impact of frost on SW radiation, we have also added Figure S5 in the supplementary material.

All the radiation data have been improved and complemented when required by corrected ERA5 values. In the version 1.1 of our data file, we have also added flags so that the user knows which data are from CNR4 and which ones are corrected ERA5 values.

We have also added details to section 3.8 on precipitation and in particular mention that "we estimate the error on the precipitation data provided to be around 20%" line 349. We believe this is useful since many snow modelers use snow height as a validation variable.

Other modifications were done to address the Reviewers' constructive suggestions, and these are detailed below.

In conclusion, we have done our best to significantly improve the paper beyond the Reviewers' requests and in particular now provide a complete and very carefully checked and discussed set of radiation time series for SW↓, SW ↑, LW↓ and LW ↑.

**Response to Reviewer 1.**

*Our responses are in blue italics, embedded into the reviewer's comments.*

The authors introduce 6 years of meteorological, soil and, snow data collected at a tundra site in the Canadian high Arctic. The observed polygonal tundra environment located on Bylot Island, Nunavut is described as well-drained, void of standing vegetation, and typical of permafrost landscape. A thorough review of instrumentation, datasets, and validation at the site are presented prior to an assessment of data quality and correction of the temporal data where necessary. The authors highlight a lack of data available to drive snow physics models in herb (graminoid) tundra environments as motivation. Thin Arctic snowpack common to this area, and greater Arctic regions, consist of contrasting high density winds slab and lower-density depth hoar elements. The predominate drivers of this composition, wind compaction and upward flux of water vapour, are clearly identified by the authors as a gap in the current generation snow physics models requiring new datasets to evaluate against and improve from.

The described work is a valuable contribution and should be published in ESSD. Public release of this dataset and its associated validation are a service to the snow, climate and meteorological communities where significant effort has been required to initiate and sustain these remote measurements. I have noted instances below where I felt minor clarifications were needed on corrections applied but in general, I felt the dataset was well documented. Where internet data resources were accessed, for example the ECCC station data, date of access should be noted in the references in case of future revision. Useful lessons and anecdotes on sensor deployment, calibration and adaptation punctuate the paper and I felt it could be more impactful if these were summarized in a concise discussion near the end. A discrete guide on how to setup an Arctic site is clearly beyond scope, but some of the more significant choices, for example the use of ERA5 over the CNR4 data, warrant clear recommendations. I would hope such as summary would be a catalyst for development of similar sites, amplifying the efforts here.

*We thank Joshua King for his overall positive recommendation and carefully consider his minor recommendations below.*

Minor comments with line numbers provided:

Lines 47, 68,73 and others – Where the authors are part of the narrative should the citations not be in-line instead of in parenthesis?

*Yes indeed. This is how endnote works and it will be fixed in the final version or at type setting.*

Line 149: What does significant mean in the contact of the spatial variability described here? It would be helpful to place this in the context of another study or even a metric of the locally observed variability to constrain the use of the word significant.

*We specified "Significant spatial variability of thermal conductivity is observed, since values near the surface vary by a factor of three within a few meters." (Line 152)*

Line 165 Superscript is missing for ^3

*Thank you. This has been fixed. (Line 167)*

Line 167: Add the total number of pits completed in the sentence or point towards a table to summarize.

*We mention that "Two to seven pits were dug" Line 169.*

Line 181: It is not clear if the correction applied was a constant or varied over time.

*In fact, we did not use the CNR4 sensor, but the datalogger sensor. Out text has been corrected and details given: "Data from 2013-2014 were missing because of sensor failure but we used CR1000 data logger temperature instead. That sensor was slightly sensitive to radiation. Based on several years of simultaneous temperature measurements of the HCS2-S3-XT and CR1000 sensors, we corrected the CR1000 sensor values by adding a linear function of downwelling SW radiation whose coefficients were optimized to obtain a zero bias." Lines 183-186.*

Line 198: This internet resource needs to be cited with the data of access.

*We mentioned that it was accessed on 5 February 2021 (Line 202)*

Line 213: The gas station equation does not have its variables introduced in text with units.

*We have added "We used PV=nRT for the gas equation of state, where P is pressure, V the volume considered, n the number of moles in V, R the gas constant and T temperature." Lines 217-218*

Line 271 The Pond Inlet airport station has an observer. Were the observer comments ever used to evaluate selected precipitation phase threshold at +0.05C?

*It is in fact 0.5°C. Thank for pointing this out. We did not use this information in our initial data set. However, we plotted the air temperature during the rain events and also during the snow events, as indicated by the Pond Inlet airport data, and +0.5°C really seems like the most sensible boundary. We added in our text (line 337) "Examination of the air temperature when observations at the Pond Inlet airport (from the ECCC web site) were snow indicate that this threshold is sensible."*

Line 279: The use of time lapses appears to be key here. Was this approach contrasted against an automated or empirical approach?

*In fact, the camera was installed in summer 2015 so that we used the automatic snow gauge and satellite data to determine snow cover dates for 2013 and 2014. The procedure has been detailed in (Domine et al., 2018). For the other years, since we had a camera, we did not seek satellite data. The snow gauge gave the same snow-in and meltout dates as the camera, within one or two day. We now mention the use of satellite data for 2013 and 2014 line 359. "For 2013 and 2014, before the deployment of several time-lapse cameras in the valley, we also used satellite images to determine snow dates, as detailed in (Domine et al., 2018)." We feel there is no need to compare gauge and camera result since we never relied on the snow gauge only.*

Line 307: It was previous stated that there was no small-scale relief at the tundra site in the context of hummock-tussock formations. Is this related to the wedge structures?

*Indeed, we stated line 128 that "There is no small-scale topography within the polygon (Figure 1) and in particular no hummock or tussock." In line 429 we stated "However, snow depth is highly spatially variable because of the small-scale relief in the ice-wedge polygon terrain." We realize this may be confusing and we now write*

*"However, snow depth is highly spatially variable because of the relief at the 10 to 20 m scale in the ice-wedge polygon terrain." Line 429.*

Line 395: This reference appears to be incomplete. Please expand on what type of correction was applied.

*There is in fact no citation possible because the relevant work by Fourteau et al. has been submitted to J. Glaciol. and submitted papers cannot be cited. We explain lines 504 that "the amount of correction decreases with increasing snow density and is about 1.1 for dense wind slabs and 1.5 for soft depth hoar." Fourteau et al. propose density-based corrections. In any case, the fact that we cannot detail this now has no impact on our current paper because we state line 505 "Data presented are uncorrected." We believe that no change on this point is warranted until the paper by Fourteau et al. is accepted.*

**Response to Reviewer 2**

*Our responses are in blue italics, embedded into the reviewer's comments.*

**\* general comments**

The authors describe and provide an extensive set of in-situ meteorological, snow and soil data to force and evaluate snow schemes in a tundra environment. Such an extensive set of forcing and evaluation data, especially for snow, is unprecedented in the litterature in this type of environment. As the most complex snow models to date fail to represent tundra snow charateristics, despite their huge significance at the global scale and w/r to global warming, this paper and dataset are in my opinion an important contribution to snow science.

*Thank you for this encouraging appreciation.*

The site and the dataset are well described, the quality assessment of the data is thorough, and the corrections performed to the data are well explained and justified (see just some minor clarification needs in the specific comments). I recommand the paper for publication in ESSD after these minor comments have been addressed.

*Thank you for this positive evaluation.*

**\* specific comments**

L51-52: "The explanation proposed (Domine et al., 2019;Domine et al., 2016b) is that Crocus and SNOWPACK were designed primarily for avalanche forecasting in the Alps, i.e. for mid latitude warm thick snowpacks while the Arctic features cold thin snowpacks." It would seem more faithful to me to cite also the other applications of these models besides avalanche forecasting (something like " were designed for avalanche forecasting and process-studies/other applications in the Alps")

*Indeed, we agree. To address this comment concisely, we rephrased our sentence "Crocus and SNOWPACK were calibrated using data from sites in the Alps", line 52, with no mention of the objectives.*

- L58-59 : "This process is not simulated by Crocus or SNOWPACK, leading to erroneous outputs." I think the attempts to partially account for this process in both models (Touzeau et al., 2018 and Jafari et al., 2020), should be mentioned here. Touzeau, A., Landais, A., Morin, S., Arnaud, L., and Picard, G. (2018). Numerical experiments on vapor diffusion in polar snow and firn and its impact on isotopes using the multi-layer energy balance model crocus in surfex v8.0. Geosci. Model Dev. 11, 2393–2418. doi: 10.5194/gmd-11-2393- 2018

Jafari M, Gouttevin I, Couttet M, Wever N, Michel A, Sharma V, Rossmann L, Maass N, Nicolaus M and Lehning M (2020) The Impact of Diffusive Water Vapor Transport on Snow Profiles in Deep and Shallow Snow Covers and on Sea Ice. Front. Earth Sci. 8:249. doi: 10.3389/feart.2020.00249

*This is true, both these papers attempt to include water vapor diffusion in Crocus and SNOWPACK. However, they do not model the water vapor transport process involved in Arctic snow, which is convection. Convection is at least an order of magnitude more efficient at transporting water vapor than diffusion, as already indicated decades ago by (Benson and Trabant, 1973) and confirmed by studies in Alaska (Johnson et al., 1987; Sturm and Benson, 1997) and at Bylot Island (Domine et al., 2016). (Touzeau et al., 2018) do a fine job at modeling diffusion in firn on ice sheets but "focus on the movement of water isotopes in the vapor in the porosity, in the absence of macroscopic air movement." (page 2394, 2nd column, 2nd paragraph). Their work is therefore not relevant to Artic snow and will not help Crocus performance in the Arctic. Their objective in any case was for isotopic exchange on ice sheets and not mass distribution in Arctic snow. The work of (Jafari et al., 2020) details water vapor diffusion in Arctic snow but to obtain a significant mass transfer these authors have to use water vapor diffusion coefficients in snow that are significantly greater than in free air. Latest work of the subject demonstrate that water vapor diffusion in snow is lower than in free air (Fourteau et al., 2021), although we admit this is a controversial topic. Furthermore, (Jafari et al., 2020) state "We acknowledge that vapor transport by diffusion may in some snow covers— such as in thin tundra snow—be small compared to convective transport, which will have to be addressed in future work.", in their abstract. In any case, we feel it would be misleading to suggest that Crocus or SNOWPACK are able to simulate water vapor*

*transport in Arctic snow. To avoid any misunderstanding, we stress that convection is the main process responsible for water vapor transport in Arctic snow and cite references on the subject. "In the thin Arctic snowpack, this process is negligible and the main process involved in determining the evolution of the density profile after precipitation and wind compaction is the upward flux of water vapor, driven mostly by convection within the snowpack (Trabant and Benson, 1972; Sturm and Johnson, 1991; Sturm and Benson, 1997; Domine et al., 2016)." Lines 54-56.*

- L181 : "Based on several years of simultaneous temperature measurements of the HCS2-S3-XT and CNR4 sensors, we corrected the CNR4 sensor values. We found that there was no bias between the two temperature measurements and a RMSD=0.784°C" I am rejoining the comment from Referee #1 on this : Details on the correction (does it involve radiations ?) would be welcome as an help/ a reference for other people encountering similar problems at their sites

*As stated in our response to Reviewer 1, our initial text was in error and we used the data logger temperature sensor. Radiation was indeed a variable in the correction as now detailed lines 185-186.*

-L332 : "Figure 2 may also indicate a decrease in summer air temperature" The statement is too vague, please clarify.

*It is indeed too vague. Discussing a temperature trend based on just 6 years of data is in fact not very meaningful. We removed this statement. In fact, we removed the section which was dedicated to snow surface temperature measured by an IR120 sensor. Now that the CNR4 problem has been solved, we replaced the section by another one "4.2 Long-wave upwelling radiation and surface temperature" based on both IR120 and CNR4 data. Lines 396 to 415.*

- L345: At several places like this one "However, all these criteria are not 100% certain, and there may be some data in the absence of snow." the honesty of the authors is much appreciated !

*Thank you. We feel it is important to state clearly the limits of our data set. We have also detailed the limits of our precipitation data and mention on 20% uncertainty on that variable (line 349).*

- L358 : "The deepest sensor was placed just above the frozen soil layer" Maybe it would be more clear if you specify that this is the frozen layer at its summer position.

*We replace this with "The deepest sensor was placed at the base of the summer thawed layer" Line 467.*

- L 395 : "Briefly, the amount of correction decreases with increasing snow density and is about 1.1 for dense wind slabs and 1.5 for soft depth hoar" Is this a multiplicative factor ? Thanks in advance for clarifying this

*Yes, measured values have to be multiplied by a factor between 1.1 and 1.5, increasing with decreasing snow density. This has been reworded: "the amount of correction decreases with increasing snow density and a multiplicative factor of about 1.1 for dense wind slabs and 1.5 for soft depth hoar must be applied." Lines 504-505.*

- Dataset : As a suggestion for later updates of this dataset, a time-series flag for each meteorological variable indicating whether the data has been gapfilled, corrected, or is the raw measure, could be usefull and more precise/exhaustive than what is currently written in the paper (for wind speed for instance)

*We agree 100% and apologize for not having done that initially. We have added a flag to the wind speed data (version 1.1) to specified gap-filled valued. We have also added flags to the radiation data.*

**References**

Benson, C. S. and Trabant, D.: Field Measurements on the flux of water vapour through dry snow, in: Proceedings of the International Symposium on The Role of Snow and Ice in Hydrology, , 1973Banff, 6-13 September 1972 1973, 291-298, 1973.

Domine, F., Barrere, M., and Sarrazin, D.: Seasonal evolution of the effective thermal conductivity of the snow and the soil in high Arctic herb tundra at Bylot Island, Canada, The Cryosphere, 10, 2573-2588, 2016.

Domine, F., Gauthier, G., Vionnet, V., Fauteux, D., Dumont, M., and Barrere, M.: Snow physical properties may be a significant determinant of lemming population dynamics in the high Arctic, Arctic Science, 4, 813-826, 2018.

Fourteau, K., Domine, F., and Hagenmuller, P.: Macroscopic water vapor diffusion is not enhanced in snow, The Cryosphere, 15, 389-406, 2021.

Jafari, M., Gouttevin, I., Couttet, M., Wever, N., Michel, A., Sharma, V., Rossmann, L., Maass, N., Nicolaus, M., and Lehning, M.: The Impact of Diffusive Water Vapor Transport on Snow Profiles in Deep and Shallow Snow Covers and on Sea Ice, Frontiers in Earth Science, 8, 2020.

Johnson, J. B., Sturm, M., Perovich, D. K., and Benson, C.: Field observations of thermal convection in a subarctic snow cover. In: Avalanche Formation, Movement and Effects (Proceedings of a Symposium held at Davos, September 1986) IAHS pub. 162, Salm, B. and Gubler, H. (Eds.), IAHS, 1987.

Sturm, M. and Benson, C. S.: Vapor transport, grain growth and depth-hoar development in the subarctic snow, J. Glaciol., 43, 42-59, 1997.

Sturm, M. and Johnson, J. B.: Natural-convection in the sub-arctic snow cover, Journal of Geophysical Research-Solid Earth and Planets, 96, 11657-11671, 1991.

Touzeau, A., Landais, A., Morin, S., Arnaud, L., and Picard, G.: Numerical experiments on vapor diffusion in polar snow and firn and its impact on isotopes using the multi-layer energy balance model Crocus in SURFEX v8.0, Geoscientific Model Development, 11, 2393-2418, 2018.

Trabant, D. and Benson, C. S.: Field experiments on the development of depth hoar, Geol. Soc. Am. Mem., 135, 309-322, 1972.